# Universality of Group Convolutional Neural Networks Based on Ridgelet Analysis on Groups

**Sho Sonoda**
RIKEN AIP
sho.sonoda@riken.jp

**Isao Ishikawa**
Ehime University
ishikawa.isao.zx@ehime-u.ac.jp

**Masahiro Ikeda**
RIKEN AIP
masahiro.ikeda@riken.jp

## Abstract

We show the universality of depth-2 group convolutional neural networks (GC-NNs) in a unified and constructive manner based on the ridgelet theory. Despite widespread use in applications, the approximation property of (G)CNNs has not been well investigated. The universality of (G)CNNs has been shown since the late 2010s. Yet, our understanding on how (G)CNNs represent functions is incomplete because the past universality theorems have been shown in a case-by-case manner by manually/carefully assigning the network parameters depending on the variety of convolution layers, and in an indirect manner by converting/modifying the (G)CNNs into other universal approximators such as invariant polynomials and fully-connected networks. In this study, we formulate a versatile depth-2 continuous GCNN $S[\gamma]$ as a nonlinear mapping between group representations, and directly obtain an analysis operator, called the *ridgelet trasform*, that maps a given function $f$ to the network parameter $\gamma$ so that $S[\gamma] = f$. The proposed GCNN covers typical GCNNs such as the cyclic convolution on multi-channel images, networks on permutation-invariant inputs (Deep Sets), and E($n$)-equivariant networks. The closed-form expression of the ridgelet transform can describe how the network parameters are organized to represent a function. While it has been known only for fully-connected networks, this study is the first to obtain the *ridgelet transform for GCNNs*. By discretizing the closed-form expression, we can systematically generate a constructive proof of the *cc-universality of finite GCNNs*. In other words, our universality proofs are more unified and constructive than previous proofs.

## 1 Introduction

In the research field of geometric deep learning [1], *group convolutional neural networks (GCNNs)* have been developed to capture the inductive bias behind a variety of datasets such as sets and point clouds [2, 3], graphs [4, 5], manifolds, groups, and homogeneous spaces [6–8, 4]. Despite the rapid growth of diversity, the *approximation property* of CNNs is less investigated than that of fully-connected neural networks (FNNs). To this date, several authors have shown the universality of (G)CNNs. That is, they can approximate some class of continuous maps with any precision [9–16]. These studies are still limited because the proofs are shown (1) in a case-by-case manner by manually assigning the parameters for a network to approximate a given function $f$, which means that once the network architecture is modified, then we need to reassign the parameters from scratch, and (2) in an indirect manner by converting/modifying the (G)CNNs into other universal approximators such as invariant polynomials and FNNs, which means that we know only indirectly about (G)CNNs.

The approximation property of *FNNs* has been investigated in the 1990s, with gradually increasing the resolution of proofs from abstract to concrete, starting from purely *existential* proofs based on the Hahn-Banach theorem [17] and the Stone-Weierstrass theorem [18], *indirect* proofs based on the Fourier transform [19, 20], the Radon transform [21, 22], $B$-splines [23, 24], to more *constructive*

proofs based on the integral representation [25], ridge functions [26], and the *ridgelet transform* [27–29]. For deep-ReLU-FNNs, further approximation properties have been investigated [30–32] in the 2010s. In this context, (G)CNN studies are at the stage of case-by-case and indirect proofs. (See § 6.1 for more details).

In this study, we show the universality of depth-2 GCNNs by devising a general notion of group convolution and developing the *ridgelet transform for GCNNs*—an analysis operator that maps a given function $f$ to the weight parameter $\gamma$ in a single hidden layer of a neural network. Consequently, our universality proof is more *unified and constructive* because our GCNN covers a wide range of typical GCNNs, and the ridgelet transform can describe how to assign the network parameters.

In the following, we describe the formulation of GCNNs to overview our main contributions.

**A Typical Convolution Layer for Images.** Given an $m_1 \times m_2$-dimensional $n_{in}$-channel input image $\boldsymbol{x} \in \mathbb{R}^{m_1 \times m_2 \times n_{in}}$, a typical *convolution layer* with $w_1 \times w_2$-dimensional $n_{in} \times n_{out}$-channel filter $\boldsymbol{a} \in \mathbb{R}^{w_1 \times w_2 \times n_{in} \times n_{out}}$ and $n_{out}$-channel bias $\boldsymbol{b} \in \mathbb{R}^{n_{out}}$ followed by an elementwise activation function $\sigma : \mathbb{R} \to \mathbb{R}$ and the aggregation with output coefficients $\boldsymbol{c} \in \mathbb{R}^{n_{out}}$ is given by

$$S[\boldsymbol{a}, \boldsymbol{b}, \boldsymbol{c}](\boldsymbol{x})(i, j) = \sum_{\ell=1}^{n_{out}} c^\ell \sigma \left( \sum_{k=1}^{n_{in}} \sum_{p=1}^{w_1} \sum_{q=1}^{w_2} a_{pq}^{k\ell} x_{i+p,j+q}^k - b^\ell \right) \tag{1}$$

for each pixel at $(i, j) \in [m_1 - w_1 + 1] \times [m_2 - w_2 + 1]$.

For technical reasons, we assume that the output channels (indexed by $\ell \in [n_{out}]$) are aggregated soon after the activation function, which may be slightly different from an ordinary formulation of CNNs, but we can understand this as a part of the subsequent layer.

In the standard formulation of GCNNs, a multi-channel image is understood as a vector-valued function on a group $G$ or a homogeneous space $G/H$, such as a product group $G = \mathbb{Z}_{m_1} \times \mathbb{Z}_{m_2}$ of cyclic groups $\mathbb{Z}_{m_i} := \mathbb{Z}/m_i\mathbb{Z}$ ($i = 1, 2$). (More geometrically, Cohen and Welling [33] phrased it as 'a section of a fiber bundle'). The convolution in the pixel directions $(i, j)$ is reformulated as a group convolution with respect to the product group, and the inner product in the channel direction $k$ is understood as the convolution with respect to the trivial action of $G$ on a 'fiber' $\mathbb{R}^{n_{in}}$.

**The Integral Representation $S[\gamma]$ of Group Convolution Layer.** Let $G$ be an *arbitrary* group, $\sigma : \mathbb{R} \to \mathbb{R}$ be an *arbitrary* nonlinear function, $\mathcal{X}$ be an *arbitrary* Hilbert space of feature vector $x$ and filter $a$, and $\gamma : \mathcal{X} \times \mathbb{R} \to \mathbb{C}$ be an *arbitrary* function, called the *parameter distribution*. We formulate a group convolution layer in an integral form, called the *integral representation*, as

$$S[\gamma](x)(g) := \int_{\mathcal{X} \times \mathbb{R}} \gamma(a, b) \sigma\big((a * x)(g) - b\big) \mathrm{d}a \mathrm{d}b, \quad x \in \mathcal{X}, \ g \in G. \tag{2}$$

This is an infinite-dimensional reparametrization of a depth-2 GCNN; namely, each function $x \mapsto \sigma((a * x)(g) - b)$ represents a single convolutional neuron, or a feature map of input $x$ parametrized by $(a, b)$, the integration over $(a, b)$ means that all the neurons are assigned, and a single function $\gamma$—the *parameter distribution*—parameterizes the assignment of each parameters $(a, b)$. Hence, $S[\gamma]$ can be understood as a *continuous neural network*. We note that, however, if we put $\gamma$ as a finite sum of Dirac's measures such as $\gamma_n := \sum_{\ell=1}^{n} c^\ell \delta_{(a^\ell, b^\ell)}$, then the integral representation can also represent a finite model

$$S[\gamma_n](x)(g) = \sum_{\ell=1}^{n} c^\ell \sigma((a^\ell * x)(g) - b^\ell), \quad x \in \mathcal{X}, \ g \in G. \tag{3}$$

In summary, $S[\gamma]$ is a mathematical model of shallow neural networks with *any* width ranging from finite to continuous. In particular, the sparsity/low-rankness of parameters are reflected as the localization/concentration of parameter distribution $\gamma$.

An advantage to use the integral representation is the *linearization trick*. Whereas a finite network $S[\gamma_n]$ is *nonlinear* in the original parameters $(a, b)$, the integral representation $S[\gamma]$ is *linear* in the parameter distribution $\gamma$. It is first emerged in the 1990s to investigate the expressive power of infinitely-wide shallow FNNs [19–22, 25, 27–29]; and it is as well common in today's deep learning theory, for example, to investigate the learning dynamics of SGD such as neural tangent kernel (NTK) [34–36], lazy learning [37], lottery tickets [38], mean field theory [39–43], and Langevin dynamics [44].

**The Ridgelet Transform** $R[f; \rho](a, b)$ is a right inverse (or pseudo-inverse) operator of the integral representation operator $S$. As an outcome of this study, we have obtained its closed-form expression:

$$R[f; \rho](a, b) := \int_{\mathcal{X}} f(x)(e)\overline{\rho(\langle a, x \rangle_{\mathcal{X}} - b)}\mathrm{d}a\mathrm{d}b, \quad (a, b) \in \mathcal{X} \times \mathbb{R}, \tag{4}$$

where $f : \mathcal{X} \to \mathbb{C}^G$ is a target vector-valued nonlinear function to be approximated, called a *feature map*, $e \in G$ is the identity element, and $\rho : \mathbb{R} \to \mathbb{C}$ is an auxiliary function, called the *ridgelet function*. Provided that $f$ is *group equivariant*, then under mild regularity assumptions, it satisfies the *reconstruction formula*

$$S[R[f; \rho]] = (\!(\sigma, \rho)\!)f, \tag{5}$$

where $(\!(\cdot, \cdot)\!)$ denote a scalar product of $\sigma$ and $\rho$. Therefore, as long as the product $(\!(\sigma, \rho)\!)$ is neither 0 nor $\infty$, we can normalize $\rho$ to satisfy $(\!(\sigma, \rho)\!) = 1$ so that $S[R[f; \rho]] = f$.

In other words, $R$ and $S$ are *analysis and synthesis operators*, and thus play the same roles as the Fourier ($F$) and inverse Fourier ($F^{-1}$) transforms, respectively. Particularly, the reconstruction formula $S[R[f; \rho]] = (\!(\sigma, \rho)\!)f$ corresponds to the Fourier inversion formula $F^{-1}[F[f]] = f$.

An advantage of the ridgelet transform is the *closed-form expression*. Despite the common belief that neural network parameters are a blackbox, the closed-form expression can clearly describe how the network parameters are organized. Previous studies on the CNN universality have also provided several construction algorithms of parameters, but these are only *particular* solutions for a CNN to represent a target function $f$, and not necessary related to, for example, deep learning solutions. For *FNNs*, on the other hand, Sonoda et al. [45] have shown that any parameter distribution $\gamma$ satisfying $S[\gamma] = f$ can always be represented as (not always single but) a linear combination of ridgelet transforms, and they [46] have shown that finite networks trained by regularized empirical risk minimization (RERM) converges to a certain unique ridgelet transform. (We note that NTK and the Gibbs distribution can also describe the parameter distribution, but NTK is limited to the kernel regime, and the Gibbs distribution is given only implicitly.) As an application, Savarese et al. [47] and their followers [48–50] have established the *representer theorems* for ReLU-FNNs by using the ridgelet transform. Although the parallel results for CNNs have not yet been published, we anticipate that the ridgelet transform could facilitate our understanding of deep learning solutions.

**Challenges and Contributions.** The closed-form expression of the ridgelet transform has been known only for FNNs, which was discovered in the 1990s independently by Murata [27], Candès [28] and Rubin [29]. (We refer to [51–53] for ridgelet analysis in the 2000s, and [54–56] for more recent results.) One of the difficulties to obtain the ridgelet transform for CNNs is that there is no unique way to formulate an "integral representation of CNNs". We note that some authors claim the "equivalence of CNNs and FNNs" (see e.g. [13]), but it is somewhat misleading because such an equivalence holds only when both CNNs and FNNs are very carefully designed. While FNNs are defined on the Euclidean space $\mathbb{R}^m$, GCNNs are defined on a more abstract space $\mathcal{X}$. For example, since the convolution on the Euclidean space can be written using Töplitz matrices, one could consider a formulation such as $\int_{\mathbb{R}^{k \times m} \times \mathbb{R}^k} \gamma(A, b)\sigma(Ax - b)\mathrm{d}A\mathrm{d}b$ where the parameter $A$ is an $k \times m$-matrix. However, this only leads to another ridgelet transform that covers less symmetries $G$. In fact, it is a version of the so-called $k$-plane ridgelet transform developed in the 2000s [51].

To circumvent this difficulty, we formulate GCNNs as general as possible by dealing with the feature space $\mathcal{X}$, group $G$, and representation $T$ in a coordinate-free manner. Eventually, we have shown the reconstruction formula for a wide range of GCNNs (as displayed in § 5), with a relatively *simple* proof. This study is the first to obtain the ridgelet transform for a general class of GCNNs. As an application, we show the *cc*-universality of GCNNs for a general class of group equivalent continuous vector-valued functions in a unified and constructive manner.

## 2 Notation and Basic Terminologies

**Notation.** For any integer $n > 0$, $[n]$ denotes the set $\{1, \ldots, n\}$. For any sets $G$ and $\mathbb{K}$, $\mathbb{K}^G$ denotes the collection $\{G \to \mathbb{K}\}$ of all mappings from $G$ to $\mathbb{K}$. For any topological space $X$, $C(X)$ and $C_c(X)$ denote the collections of all continuous functions on $X$, and continuous functions on $X$ with compact support, respectively. We note that when $X$ is compact, then $C(X) = C_c(X)$. For any measure space $X$ and number $p \in [1, \infty]$, $L^p(X)$ denotes the space of $p$-integrable functions on $X$.

## 2.1 Fourier Analysis on $\mathbb{R}^d$

We refer to [57, 58, 54] for more details on Fourier transform and tempered distributions ($\mathcal{S}'$).

**Schwartz Distributions.** For any integer $d > 0$, $\mathcal{S}(\mathbb{R}^d)$ and $\mathcal{S}'(\mathbb{R}^d)$ denote the classes of Schwartz test functions (or rapidly decreasing functions) and tempered distributions on $\mathbb{R}^d$, respectively. Namely, $\mathcal{S}'$ is the topological dual of $\mathcal{S}$. In this study, $\mathcal{S}'(\mathbb{R})$ and $\mathcal{S}(\mathbb{R})$ are assigned as classes of activation and ridgelet functions, respectively. We note that $\mathcal{S}'(\mathbb{R})$ includes truncated power functions $\sigma(b) = b_+^k = \max\{b, 0\}^k$ such as step function for $k = 0$ and ReLU for $k = 1$.

**Fourier Transform.** The Fourier transform on the Euclidean space $\mathbb{R}^d$ and its inversion formula has been defined on (at least) three different function classes: $L^1(\mathbb{R}^d), L^2(\mathbb{R}^d)$ and $\mathcal{S}'(\mathbb{R}^d)$. When $f \in L^1(\mathbb{R}^d)$ and $\widehat{f} \in L^1(\mathbb{R}^d)$, the inversion formula holds "at every continuous point $\boldsymbol{x}$ of $f$", which is a pointwise equation. When $f \in L^2(\mathbb{R}^d)$, the inversion formula holds "in $L^2$", which is not a pointwise equation because the equation "$f = g$ in $L^2$" is defined as "$f(\boldsymbol{x}) = g(\boldsymbol{x})$ a.e.". Similarly, when $f \in \mathcal{S}'(\mathbb{R}^m)$, the inversion formula holds "in $\mathcal{S}'$". We use the third definition for computing the Fourier transform of activation functions $\sigma \in \mathcal{S}'(\mathbb{R})$ such as ReLU and $\tanh$.

## 2.2 Group Representation

Let $G$ be a group, let $\mathcal{X}$ be a vector space over a field $\mathbb{K}$, and let $GL(\mathcal{X})$ be the general linear group on $\mathcal{X}$. A *group representation $T$* of the group $G$ on the vector space $\mathcal{X}$ is a group homomorphism from $G$ to $GL(\mathcal{X})$, that is, a map $T : G \to GL(\mathcal{X}); g \mapsto T_g$ satisfying $T_{gh} = T_g T_h$ for all $g, h \in G$. When $G$ is a topological group, we further assume that the action $G \times \mathcal{X} \to \mathcal{X}; (g, x) \mapsto T_g[x]$ be continuous. Here, $\mathcal{X}$ is called the *representation space*. We refer to [59] for more details on group representation.

**Regular Representation.** Let $\mathcal{X}$ be the vector space of all functions on $G$, i.e., $\mathcal{X} = \mathbb{K}^G$. The *(left) regular representation $L$* is a group representation defined on $\mathcal{X}$ as

$$L_g[x](h) := x(g^{-1}h), \quad g, h \in G, \ x \in \mathcal{X} = \mathbb{K}^G. \tag{6}$$

In particular, when $G$ is a locally compact Haussdorf (LCH) group, then it has a (left) invariant measure $\mu$, and we can define the collection $L^2(G)$ of all square integrable functions on $G$ with respect to the canonical inner product $\langle x, y \rangle_{L^2(G)} := \int_G x(g)\overline{y(g)}\mathrm{d}\mu(g)$ for any measurable functions $x, y : G \to \mathbb{C}$. It is known that the regular representation on $\mathcal{X} = L^2(G)$ is a unitary representation.

**Dual Representation.** For any group representation $T : G \to GL(\mathcal{X})$, the *dual representation* $T^* : G \to GL(\mathcal{X}')$ is a group representation defined on the dual vector space $\mathcal{X}'$ as the transpose of $T_{g^{-1}}$, that is, $T_g^* = T_{g^{-1}}^\top$. When $\mathcal{X}$ is a Hilbert space with inner product $\langle \cdot, \cdot \rangle_\mathcal{X}$, then it satisfies the following relation:

$$\langle T_g[x], y \rangle_\mathcal{X} = \langle x, T_{g^{-1}}^*[y] \rangle_\mathcal{X}, \quad g \in G, \ x, y \in \mathcal{X}. \tag{7}$$

**Matrix Element.** For any group representation $T : G \to GL(\mathcal{X})$, the *matrix element* (or the *matrix coefficient*) of $T$ is a bilinear functional $f_{a,x}$ on $G$ defined by

$$f_{a,x}(g) := a[T_g[x]], \quad g \in G, \ x \in \mathcal{X}, \ a \in \mathcal{X}' \tag{8}$$

where $x$ is a vector in $\mathcal{X}$ and $a \in \mathcal{X}'$ is a continuous linear functional on $\mathcal{X}$. When $\mathcal{X}$ is a Hilbert space, then (identifying $\mathcal{X}'$ with $\mathcal{X}$) it can be written as

$$f_{a,x}(g) = \langle T_g[x], a \rangle_\mathcal{X}, \quad g \in G, \ a, x \in \mathcal{X}. \tag{9}$$

In the next section, we use this quantity as the generalized form of the *group convolution*.

## 2.3 Universality

The notion of *universality* in machine learning can be rephrased as the *density* in mathematics, and thus it has several definitions. (See, e.g., [60, 61]). In this study, we show the so-called *cc-univesality*, one of the standard universalities in the machine learning theory.

*cc*-**Universality.** Let $X$ be a topological space, and let $\mathtt{NN}$ be a collection of functions (e.g., neural networks) on $X$. The *cc*-univesality of $\mathtt{NN}$ is defined as the density of $\mathtt{NN}$ in $C(X)$ endowed with the topology of *compact convergence*, that is, for any compact subset $K \subset X$, continuous function $f \in C(K)$, and all $\varepsilon > 0$, there exists a function $g \in \mathtt{NN}$ such that

$$\|f - g|_K\|_{C(K)} := \sup_{x \in K} |f(x) - g(x)| < \varepsilon, \tag{10}$$

where $g|_K$ denotes the restriction of $g$ to $K$.

# 3 Functions on Abstract Hilbert Space $\mathcal{X}$

We introduce an extended group convolution on $\mathcal{X}$, a uniform norm and the group-equivariance for functions on $\mathcal{X}$, an induced measure and an induced Fourier transform on $\mathcal{X}$, and a projection to $\mathcal{X}_m$.

## 3.1 $(G, T)$-**Convolution** $*_T : \mathcal{X} \times \mathcal{X} \to \mathbb{C}^G$

**Definition 1.** Let $G$ be a group, let $\mathcal{X}$ be a Hilbert space with inner product $\langle \cdot, \cdot \rangle_{\mathcal{X}}$ over a field $\mathbb{K}$, and let $T : G \to GL(\mathcal{X})$ be a representation of $G$ on $\mathcal{X}$. For any $a, x \in \mathcal{X}$ and $g \in G$, we define the $(G, T)$-*convolution* as

$$(a *_T x)(g) := \langle x, T_g^*[a] \rangle_{\mathcal{X}} = \langle T_{g^{-1}}[x], a \rangle_{\mathcal{X}}. \tag{11}$$

We remark (1) that this is simply a paraphrase of the *matrix element of a group representation* (see the previous section), and (2) that this is *not necessarily* a binary operation because $\mathcal{X} \neq \mathbb{C}^G$ in general. Nevertheless, we call it a *convolution* simply because it covers a wide range of 'group convolutions' in today's GCNN literature.

**Example 1.** An orthodox group convolution is reproduced when $T$ is the regular representation (of a LCH group $G$) on $\mathcal{X} = L^2(G)$, i.e., $T_g^*[a](h) = a(g^{-1}h)$. In fact,

$$\langle x, T_g^*[a] \rangle_{L^2(G)} = \int_G x(h)\overline{a(g^{-1}h)}\mathrm{d}\mu(h) = \int_G x(h)\tilde{a}(h^{-1}g)\mathrm{d}\mu(h) = (x *_G \tilde{a})(g), \tag{12}$$

where $\tilde{a}(g) := \overline{a(g^{-1})}$ is an involution.

**Example 2.** The cyclic convolution for an $n$-channel image $\boldsymbol{x} = (x_{ij}^k) \in \mathbb{R}^{m_1 \times m_2 \times n}$ is understood as the case when $G = \mathbb{Z}_{m_1} \times \mathbb{Z}_{m_2}$, $\mathcal{X} = \mathbb{R}^{m_1 \times m_2 \times n}$, and $T_{(p,q)}^*[\boldsymbol{a}](i, j, k) = a_{i-p,j-q}^k$, then

$$\langle \boldsymbol{x}, T_{(p,q)}^*[\boldsymbol{a}] \rangle_{\mathbb{R}^{m_1 \times m_2 \times n}} = \sum_{i,j,k} x_{ij}^k a_{i-p,j-q}^k. \tag{13}$$

While the post-activation feature $\sigma((a * x)(g) - b)$ is a function on $G$, the input feature $x$ can be an arbitrary abstract vector, which is more general than typical GCNN formulations where feature $x$ is supposed to be a vector-valued function on $G$ or $G/H$. This is an advantage for a more geometric understanding of CNNs, since the theory becomes free from the specification of $x$.

## 3.2 Continuous $(G, T)$-**Equivariant Vector-Valued Function** $f : \mathcal{X} \to C(G)$

**Definition 2.** We say a vector-valued function $f : \mathcal{X} \to \mathbb{C}^G$ is $(G, T)$-*equivariant* when

$$f(T_g[x])(h) = L_g[f(x)](h) = f(x)(g^{-1}h), \quad x \in \mathcal{X}, \ g, h \in G. \tag{14}$$

Here, we restrict the definition for a special case of the regular representation $L$. This is simply due to the fact that our GCNN satisfies this case.

**Definition 3.** Let $G$ be a topological group. For any vector-valued function $f : \mathcal{X} \to \mathbb{C}^G$, put

$$\|f\|_{C(\mathcal{X};C(G))} := \|f\|_{C(\mathcal{X}) \to C(G)} := \sup_{x \in \mathcal{X}} \left| \sup_{g \in G} |f(x)(g)| \right|. \tag{15}$$

By $C_{equi}(\mathcal{X}; C(G))$, we denote the normed vector space of all *continuous* $(G, T)$-*equivariant* $C(G)$-*valued* functions on $\mathcal{X}$ equipped with the uniform norm $\| \cdot \|_{C(\mathcal{X};C(G))}$.

We note that the topology of uniform norm $\| \cdot \|_{C(\mathcal{X};C(G))}$ is stronger than the topology of compact convergence, which is employed in the *cc*-universality argument. In fact, $C_{equi}(\mathcal{X}; C(G))$ need not be complete (or Banach) to show the *cc*-universality.

## 3.3 Induced Lebesgue Measure $\lambda$ and Induced Fourier Transform $\widehat{\cdot}$ on Subspace $\mathcal{X}_m$

Let $\mathcal{X}_m$ denote an $m$-dimensional subspace of $\mathcal{X}$, and let $\{e_i\}_{i\in[m]}$ be an orthonormal basis of $\mathcal{X}_m$.

**Induced Lebesgue Measure on $\mathcal{X}_m$.** We induce the Lebesgue measure $\lambda$ on $\mathcal{X}_m$ by pushing forward the Lebesgue measure $d\boldsymbol{x}$ on $\mathbb{R}^m$ via an isometric linear embedding $\phi : \mathbb{R}^m \to \mathcal{X}_m$. For example, take a linear embedding $\phi(\boldsymbol{x}) := \sum_{i\in[m]} x_i e_i$. Then, it preserves the length, and we can induce the Lebesgue measure $\lambda$ on $\mathcal{X}_m$ as the push forward measure $\lambda = \phi_\sharp d\boldsymbol{x}$ so that the volume of a hypercube $Q = \{\sum_{i\in[m]} c_i e_i \mid c_i \in [a_i, b_i]\}$ in $\mathcal{X}_m$ is calculated as $\lambda(Q) = \prod_{i\in[m]} |b_i - a_i|$, and the integration of a measurable function $f : \mathcal{X}_m \to \mathbb{C}$ over a measurable set $E \subset \mathcal{X}_m$ is calculated as

$$\int_E f(x)\mathrm{d}\lambda(x) = \int_{\mathbb{R}^m} 1_E\left(\sum_{i=1}^m x_i e_i\right) f\left(\sum_{i=1}^m x_i e_i\right) \prod_{i=1}^m \mathrm{d}x_i = \int_{\phi^{-1}(E)} f \circ \phi(\boldsymbol{x})\mathrm{d}\boldsymbol{x}. \qquad (16)$$

As far as there is no risk of confusion, we denote $\mathrm{d}x$ instead of $\mathrm{d}\lambda(x)$.

**Induced Fourier Transform on $\mathcal{X}_m$.** Using $\lambda$, we induce the Fourier transform on $\mathbb{R}^m$ as below: For any function $f : \mathcal{X}_m \to \mathbb{C}$,

$$\widehat{f}(y) := \int_{\mathcal{X}_m} f(x)e^{-i\langle x,y\rangle_{\mathcal{X}_m}}\mathrm{d}\lambda(x), \quad f(x) \overset{\star}{=} \frac{1}{(2\pi)^m} \int_{\mathcal{X}_m} \widehat{f}(y)e^{i\langle x,y\rangle_{\mathcal{X}_m}}\mathrm{d}\lambda(y). \qquad (17)$$

Here, the equality $\overset{\star}{=}$ holds in at least three different senses (see the comments in § 2.1).

We remark (1) that once the subspace $\mathcal{X}_m$ is fixed, the induced Fourier transform is unique up to the orthogonal transformation of the basis $\{e_i\}_{i\in[m]}$, and (2) that the induced Fourier transform "on $\mathcal{X}$" should not be confused with the Fourier transform "on group $G$". Especially, this *cannot* map a convolution $x *_T a$, an element in $\mathbb{C}^G$, to a point product such as "$\widehat{x} \cdot \widehat{a}$".

## 3.4 Projection $P : \mathcal{X} \to \mathcal{X}_m$ and Extension Operator $P^*$

In order to induce the Lebesgue measure $\lambda$, we assume that the dimension of $\mathcal{X}_m$ to be finite. As a side effect of this assumption, the image $T_G[\mathcal{X}_m] := \{T_g[x] \mid g \in G, x \in \mathcal{X}_m\}$ can extend toward the outside of $\mathcal{X}_m$; that is, $\mathcal{X}_m$ is not necessarily $G$-invariant ($T_G[\mathcal{X}_m] \subset \mathcal{X}_m$). To avoid an "undefined error" such as to input $x$ outside of $\mathcal{X}_m$ for a function $f$ defined only on $\mathcal{X}_m$, we introduce projection $P$ and extension $P^*$ as below. When $\dim \mathcal{X} < \infty$, we can omit $P$ by putting $\mathcal{X}_m = \mathcal{X}$ (so $P = \mathrm{Id}$), because by the definition of the group representation, always $T_G[\mathcal{X}] = \mathcal{X}$.

Let $\mathcal{X}_m^\perp$ denote the orthogonal complement of $\mathcal{X}_m$ in $\mathcal{X}$. Let $P : \mathcal{X} \to \mathcal{X}_m$ denote the orthogonal projection onto $\mathcal{X}_m$. For any function $f : \mathcal{X}_m \to \mathbb{C}^G$, put

$$P^*f(z)(g) := f(P(z))(g), \quad z \in \mathcal{X}, g \in G. \qquad (18)$$

This extends $f$ (on a subspace $\mathcal{X}_m$) to the entire space $\mathcal{X}$ as a constant function on $\mathcal{X}_m^\perp$; that is, $P^*f(x \oplus y) = f(x \oplus 0)$ for each $x \oplus y \in \mathcal{X}_m \oplus \mathcal{X}_m^\perp$.

# 4 Main Results

We introduce the $(G, T)$-convolutional neural networks and the corresponding ridgelet transform, and present the reconstruction formula for continuous GCNNs and the *cc*-universality for finite GCNNs.

Throughout this section, we fix a representation $T : G \to GL(\mathcal{X})$ of a group $G$ on a (potentially infinite-dimensional) Hilbert space $\mathcal{X}$ over a field $\mathbb{K}$ endowed with an inner product $\langle \cdot, \cdot \rangle_\mathcal{X}$, and fix an $m$-dimensional closed subspace $\mathcal{X}_m$ of $\mathcal{X}$ equipped with an induced Lebesgue measure $\lambda$. Let $k := \dim_\mathbb{R} \mathbb{K}$ denote the real dimension of $\mathbb{K}$, that is, $k = 1$ for $\mathbb{K} = \mathbb{R}$ and $k = 2$ for $\mathbb{K} = \mathbb{C}$. Let $e$ denote the identity element of $G$.

## 4.1 Integral Representation of $(G, T)$-Convolutional Neural Network

**Definition 4.** For any functions $\gamma : \mathcal{X}_m \times \mathbb{K} \to \mathbb{C}$ and $\sigma : \mathbb{K} \to \mathbb{C}$, we define the *integral representation of $(G, T)$-convolutional neural network* as a vector-valued function $\mathcal{X} \to \mathbb{C}^G$,

$$S[\gamma](x)(g) := \int_{\mathcal{X}_m \times \mathbb{K}} \gamma(a, b)\sigma((a *_T x)(g) - b)\mathrm{d}\lambda(a)\mathrm{d}b, \quad x \in \mathcal{X}, g \in G. \qquad (19)$$

Here, we call $\gamma$ a parameter distribution, and $\sigma$ an activation function. If there is no risk of confusion, we abbreviate $\mathrm{d}\lambda(a)$ as $\mathrm{d}a$.

It is easy to see that a $(G, T)$-CNN is $(G, T)$-equivariant. In fact, for every $g, h \in G$,

$$S[\gamma](T_g[x])(h) = \int_{\mathcal{X}_m \times \mathbb{K}} \gamma(a, b)\sigma(\langle T_{(g^{-1}h)^{-1}}[x], a\rangle_{\mathcal{X}} - b)\mathrm{d}a\mathrm{d}b = S[\gamma](x)(g^{-1}h). \qquad (20)$$

In addition, at the identity element $g = e$, it is reduced to a FNN:

$$S[\gamma](x)(e) = \int_{\mathcal{X}_m \times \mathbb{K}} \gamma(a, b)\sigma(\langle x, a\rangle_{\mathcal{X}} - b)\mathrm{d}a\mathrm{d}b, \quad x \in \mathcal{X} \qquad (21)$$

and it satisfies a *projection property*:

$$S[\gamma](P[x])(e) = S[\gamma](x)(e), \quad x \in \mathcal{X}. \qquad (22)$$

## 4.2  Ridgelet Transform and Scalar Product of Activation Function

**Definition 5.** For any functions $f : \mathcal{X}_m \to \mathbb{C}^G$ and $\rho : \mathbb{K} \to \mathbb{C}$, we define the *ridgelet transform* as

$$R[f; \rho](a, b) := \int_{\mathcal{X}_m} f(x)(e)\overline{\rho(\langle x, a\rangle_{\mathcal{X}} - b)}\mathrm{d}x, \quad (a, b) \in \mathcal{X}_m \times \mathbb{K}. \qquad (23)$$

Here $e$ denotes the identity element of $G$.

**Definition 6.** For any tempered distribution $\sigma \in \mathcal{S}'(\mathbb{K})$ and function $\rho \in \mathcal{S}(\mathbb{K})$, put a *scalar product* as

$$((\sigma, \rho)) := (2\pi)^{m-k} \int_{\mathbb{K}} \sigma^{\sharp}(\omega)\overline{\rho^{\sharp}(\omega)}|\omega|^{-m}\mathrm{d}\omega. \qquad (24)$$

Here, $\cdot^{\sharp}$ denotes the Fourier transform on $\mathbb{K}$, which is identified with the Fourier transform on $\mathbb{R}^k$ with $k = \dim_{\mathbb{R}} \mathbb{K}$. We note that $\sigma^{\sharp}$ is defined in the sense of tempered distributions.

The derivations of the ridgelet transform and the scalar product are clarified in the proof of the reconstruction formula. Some readers may notice that the ridgelet transform for GCNN is formally the same as the one for FNNs, and may wonder why inner product $\langle x, a\rangle_{\mathcal{X}}$ instead of group convolution $(a *_T x)(g)$. Indeed, this is a consequence of two facts (1) that a group convolution at the identity $e$ is reduced to an inner product: $(a *_T x)(e) = \langle x, a\rangle_{\mathcal{X}}$, and (2) that when $f$ is $(G, T)$-equivariant, then the value $f(x)(g)$ at each $g \in G$ is determined by translating the value $f(x)(e)$ at the identity.

## 4.3  Reconstruction Formula, or the Universality of Continuous GCNNs

We state the first half of our main results. For $f : \mathcal{X}_m \to \mathbb{C}^G$, we write $f_e(x) := f(x)(e)$ for short.

**Theorem 1** (Main Theorem 1/2). *Given a function $f : \mathcal{X}_m \to \mathbb{C}^G$, assume (A1) that $P^*f : \mathcal{X} \to \mathbb{C}^G$ is $(G, T)$-equivariant, i.e.,*

$$P^*f(T_g[z])(h) = f(z)(g^{-1}h), \quad \text{for every } z \in \mathcal{X} \text{ and } g, h \in G; \qquad (25)$$

*and (A2) that $f$ satisfies at least one of the following conditions: (A2a) both $f_e$ and $\widehat{f}_e$ are absolute-integrable, i.e., $f_e, \widehat{f}_e \in L^1(\mathcal{X}_m)$, (A2b) $f_e$ is square-integrable, i.e., $f_e \in L^2(\mathcal{X}_m)$, or (A2c) $f_e$ is a tempered distribution, i.e., $f_e \in \mathcal{S}'(\mathcal{X}_m)$. Then, the following reconstruction formula holds:*

$$S[R[f; \rho]](x)(g) = \int_{\mathcal{X}_m \times \mathbb{K}} R[f; \rho](a, b)\sigma((a *_T x)(g) - b)\mathrm{d}a\mathrm{d}b \overset{\star}{=} ((\sigma, \rho))f(x)(g), \qquad (26)$$

*where the equality $\overset{\star}{=}$ holds at every continuous point $x_c$ of $f$ for (A2a), in $L^2$ for (A2b), and in $\mathcal{S}'$ for (A2c), respectively.*

The proof is given in Appendix A.1.

### 4.4  $cc$-Universality of Finite GCNNs

Finally, we state the second half of our main results. Let $\mathtt{NN}$ be the collection of finite GCNNs, that is,

$$\mathtt{NN} := \bigcup_{n \in \mathbb{N}} \left\{ f_n(x)(g) = \sum_{i=1}^{n} c_i \sigma((a_i *_T x)(g) - b_i) \,\middle|\, (a_i, b_i, c_i) \in \mathcal{X}_m \times \mathbb{K} \times \mathbb{C}, i \in [n] \right\}. \quad (27)$$

Since the reconstruction formula $S[\gamma_f] = f$ with $\gamma_f = R[f; \rho]$ holds for an arbitrary function $f$, we can construct a sequence $\{f_n\}_{n \in \mathbb{N}}$ of finite $(G, T)$-CNNs that converges to an arbitrary target function $f$, namely

$$f_n \to f \quad \text{as} \quad n \to \infty, \quad (28)$$

by discretizing the continuous network $S[\gamma_f]$ and distribution $\gamma_f$ into finite sums

$$f_n(x)(g) := S[\gamma_n](x)(g) = \sum_{i=1}^{n} c_i \sigma((a_i *_T x)(g) - b_i) \quad \text{with} \quad \gamma_n := \sum_{i=1}^{n} c_i \delta_{(a_i, b_i)} \quad (29)$$

in a 'nice' manner so that $\gamma_n \to \gamma_f = R[f; \rho]$ as $n \to \infty$. This is the primitive idea behind the constructive proof of the following $cc$-universality of finite $(G, T)$-CNNs based on ridgelet analysis.

To state a regularity assumption on the activation function $\sigma$, we introduce the forward difference operator $\Delta_\theta^n$ with difference $\theta > 0$, defined as

$$\Delta_\theta^1[\sigma](t) := \sigma(t + \theta) - \sigma(t), \quad \Delta_\theta^{n+1}[\sigma](t) := \Delta_\theta^1 \circ \Delta_\theta^n[\sigma](t). \quad (30)$$

**Theorem 2** (Main Theorem 2/2). *For an activation function $\sigma \in \mathcal{S}'(\mathbb{K})$, assume (A3) that there exist $n \in \mathbb{N}$ and $\theta > 0$ such that $\Delta_\theta^n[\sigma]$ is bounded and Lipschitz continuous. Then, $\mathtt{NN}$ is cc-universal; that is, for any continuous $(G, T)$-equivariant $C(G)$-valued function $f \in C_{equi}(\mathcal{X}_m; C(G))$, and for any compact sets $K \subset \mathcal{X}_m$ and $L \subset G$, there exists a sequence $\{f_n\}_{n \in \mathbb{N}} \subset \mathtt{NN}$ of finite GCNNs satisfying*

$$\|f - f_n\|_{C(K; C(L))} = \sup_{x \in K} \sup_{g \in L} |f(x)(g) - f_n(x)(g)| \to 0, \quad n \to \infty. \quad (31)$$

The proof is given in Appendix A.2. Here, $f \in C_{equi}(\mathcal{X}_m; C(G))$ means that $P^*f$ is $(G, T)$-equivariant.

## 5  Examples

We display the ridgelet transforms and reconstruction formulas for a few typical GCNNs. Besides, we calculated in Examples 5 and 8 the ridgelet transforms of a differential filter, which is often reported to be acquired as a feature map in the first layer of deep CNNs for image recognition [62, 63].

### 5.1  Finite Periodic Convolution Layer

**Example 3** (For 1-dimensional periodic signals)**.** The periodic convolution corresponds to the case when $\mathbb{K} = \mathbb{R}$, $G = \mathbb{Z}_m \cong [m] = \{0, 1, \ldots, m - 1\}$, $\mathcal{X} = L^2(G) \cong \mathbb{R}^m$ equipped with the inner product $\langle x, y \rangle := \frac{1}{m} \sum_{i \in [m]} x_i y_i$, and $T_i[x](j) := x_{j-i}$ thus $(a *_T x)(i) = \frac{1}{m} \sum_{j \in [m]} a_j x_{i+j}$. Therefore, the ridgelet transform and the reconstruction formula are given by

$$R[f; \rho](a, b) = \int_{\mathbb{R}^m} f(\boldsymbol{x})(0) \overline{\rho\left(\tfrac{1}{m} \sum_{i \in [m]} a_i x_i - b\right)} \mathrm{d}\boldsymbol{x},$$

$$S[R[f; \rho]](x)(i) = \int_{\mathbb{R}^m \times \mathbb{R}} R[f; \rho](\boldsymbol{a}, b) \sigma\left(\tfrac{1}{m} \sum_{j \in [m]} a_j x_{i+j} - b\right) \mathrm{d}\boldsymbol{a}\mathrm{d}b = ((\sigma, \rho)) f(x)(i).$$

**Example 4** (For 2-dimensional multi-channel periodic images)**.** A 2-dimensional $n$-channel image is identified with a vector-valued function $x : \mathbb{Z}_m^2 \to \mathbb{R}^n$, thus $\mathcal{X} \cong \mathbb{R}^{m^2 \times n}$. Let $x_{ij}^k$ denote the $(i, j)$-th component in the $k$-th channel of $x \in \mathcal{X}$. Let $G = \mathbb{Z}_m^2$, and put $T_{(p,q)}[x]_{ij}^k := x_{i-p,j-q}^k$. Therefore, the ridgelet transform and the reconstruction formula are given by

$$R[f; \rho](a, b) = \int_{\mathbb{R}^{m^2 n}} f(x)(0,0) \overline{\rho\left(\tfrac{1}{m^2 n} \sum_{k \in [n]} \sum_{i,j \in [m]} a_{ij}^k x_{ij}^k - b\right)} \mathrm{d}\boldsymbol{x}.$$

$$S[R[f; \rho]](x)_{ij} = \int_{\mathbb{R}^{m^2 n} \times \mathbb{R}} R[f; \rho](a, b) \sigma\left(\tfrac{1}{m^2 n} \sum_{k \in [n]} \sum_{p,q \in [m]} a_{pq}^k x_{p+i,q+j}^k - b\right) \mathrm{d}\boldsymbol{a}\mathrm{d}b = ((\sigma, \rho)) f(x)_{ij}.$$

**Example 5** (Difference operator (with cutoff function)). A difference operator on $x : [m] \to \mathbb{R}$ is given by $x = \sum_{i \in [m]} x_i \delta_i \mapsto f(x) = \sum_{i \in [m]} (x_{i+1} - x_i) \delta_i$, which is $(G, T)$-equivariant: $f(T_k^*[x])(i) = T_k^*[x]_{i+1} - T_k^*[x]_i = x_{i+1-k} - x_{i-k} = f(x)(i - k)$. Since $f(x)(0) = x_1 - x_0$,

$$R[f|_K; \rho](a, b) = \int_{\mathbb{R}^m} (x_1 - x_0) 1_K(\boldsymbol{x}) \overline{\rho\left(\tfrac{1}{m} \sum_{i \in [m]} a_i x_i - b\right)} \mathrm{d}\boldsymbol{x}.$$

We note since $\boldsymbol{x} \mapsto x_1 - x_0$ is not integrable in $\mathbb{R}^m$, we restrict $f$ to a compact set $K \subset \mathbb{R}^m$, and impose the indicator function $1_K$ as an auxiliary cutoff function.

## 5.2 (Deep Sets) Permutation Equivariant Maps on A Finite Set

**Example 6.** Let $\mathcal{X} = \mathbb{R}^m, G \leqslant \mathrm{S}_m$ and $T_g[x] = (x_{g^{-1}(1)}, x_{g^{-1}(2)}, \ldots, x_{g^{-1}(m)})$. Thus $\langle a, x \rangle_{\mathcal{X}} = \frac{1}{m} \sum_{i \in [m]} a_i x_i$, and $(a *_T x)(g) = \frac{1}{m} \sum_{p \in [m]} a_p x_{g(p)}$. So,

$$R[f; \rho](a, b) = \int_{\mathbb{R}^m} f(\boldsymbol{x})(e) \overline{\rho\left(\tfrac{1}{m} \sum_{i \in [m]} a_i x_i - b\right)} \mathrm{d}\boldsymbol{x},$$

$$S[R[f; \rho]](x)(g) = \int_{\mathbb{R}^m \times \mathbb{R}} R[f; \rho](\boldsymbol{a}, b) \sigma\left(\tfrac{1}{m} \sum_{p \in [m]} a_p x_{g(p)} - b\right) \mathrm{d}\boldsymbol{a}\mathrm{d}b = ((\sigma, \rho)) f(x)(g).$$

## 5.3 Continuous Periodic Convolution Layer

**Example 7.** Let $G = \mathbb{T} := \mathbb{R}/2\pi\mathbb{Z} \cong \{e^{i\theta} \mid \theta \in [-\pi, \pi]\}$ be the 1-dimensional torus group, which is one of the most basic continuous group. As a consequence of the Fourier series expansion, $L^2(\mathbb{T})$ is spanned by $\{e^{in\theta} \mid n \in \mathbb{N}\}$. Hence, we can take $\mathcal{X}$ to be an $m$-dimensional subspace $\mathcal{X} := \{\sum_{|n|<m} x_n e^{in\theta} \mid x_{-n} = x_n \in \mathbb{R}\}$ equipped with an inner product $\langle x, y \rangle_{\mathcal{X}} := \int_{\mathbb{T}} x(\theta) \overline{y(\theta)} \mathrm{d}\theta$. We note that the constraint $x_n = x_{-n}$ implies $\sum_{|n|<m} x_n e^{in\theta} = \sum_{|n|<m} x_n \cos(n\theta)$ and thus any signal $x \in \mathcal{X}$ is a bandlimited real-valued continuous signal with each coefficient $x_n$ being the $n$-th frequency spectrum. Put $T_\alpha[x](\theta) := x(\theta - \alpha)$, then $(a * x)(\alpha) = \int_{\mathbb{T}} a(\theta) x(\alpha - \theta) \mathrm{d}\theta = \sum_{|n|<m} a_n x_n e^{in\alpha} = \sum_{|n|<m} a_n x_n \cos(n\alpha)$ (by the convolution theorem and the constraint). Therefore,

$$R[f; \rho](a, b) := \int_{\mathbb{R}^m} f(\boldsymbol{x})(0) \overline{\rho\left(\sum_{|n|<m} a_n x_n - b\right)} \mathrm{d}\boldsymbol{x},$$

$$S[R[f; \rho]](x)(\theta) = \int_{\mathbb{R}^m \times \mathbb{R}} R[f; \rho](\boldsymbol{a}, b) \sigma\left(\sum_{|n|<m} a_n x_n \cos(n\theta) - b\right) \mathrm{d}\boldsymbol{a}\mathrm{d}b = ((\sigma, \rho)) f(x)(\theta).$$

**Example 8** (Differential operator (with convergence factor)). A differential operator $\frac{\mathrm{d}}{\mathrm{d}\theta}$ is calculated as $x = \sum_{|n|<m} x_n e^{in\theta} \mapsto f(x) = \sum_{|n|<m} n x_n e^{in\theta}$. Since $\boldsymbol{x} \mapsto f(\boldsymbol{x})(0) = \sum_{|n|<m} n x_n$ is not integrable on $\mathbb{R}^m$, we impose a convergence factor $\phi_t$ as follows. $f_t(x)(\theta) := f(x) \phi_t(x) = \frac{\mathrm{d}}{\mathrm{d}\theta} x(\theta) \phi_t(x) = \sum_{|n|<m} n x_n \phi_t(x) e^{in\theta}$. Here, $(\phi_t)_{t>0} \subset \mathcal{S}(\mathcal{X})$ is a family of convergence factors that satisfies (1) the first moment $\int_{\mathcal{X}} |x|_{\mathcal{X}} |\phi_t(x)| \mathrm{d}x$ exists at every $t$, (2) $\phi_t \to 1$ in the weak sense as $t \to \infty$, and (3) (continuous and) $(G, T)$-equivariant. For example, we can take a Gaussian $\phi_t(x) = \exp(-|x|_{\mathcal{X}}^2/4t)$. Hence,

$$R[f_t; \rho](a, b) = \int_{\mathbb{R}^m} \sum_{|n|<m} n x_n \phi_t(\boldsymbol{x}) \overline{\rho\left(\sum_{|n|<m} a_n x_n - b\right)} \mathrm{d}\boldsymbol{x}.$$

## 5.4 Euclidean group $\mathrm{E}(n)$ equivariant map

**Example 9.** The Euclidean group $\mathrm{E}(n)$ is a semidirect product $\mathbb{R}^n \rtimes \mathrm{O}(n)$ of the translational group $\mathbb{R}^n$ and the orthogonal group $\mathrm{O}(n)$, which acts on $\mathbb{R}^n$ as $(U, s) \cdot t := Ut + s$ for any $t \in \mathbb{R}^n$ and $(U, s) \in O(n) \times \mathbb{R}^n$. So, put $\mathcal{X} \subset L^2(\mathbb{R}^n)$ and $T_{(U,s)}[x](t) := x(U^{-1}(t - s))$. Then,

$$R[f; \rho](a, b) := \int_{\mathcal{X}} f(x)(I, 0) \overline{\rho\left(\int_{\mathbb{R}^n} a(t) \overline{x(t)} \mathrm{d}t - b\right)} \mathrm{d}x$$

$$S[R[f; \rho]](x)(U, t) = \int_{\mathcal{X} \times \mathbb{R}} R[f; \rho](a, b) \sigma\left(\int_{\mathbb{R}^n} a(U^{-1}(t - s)) \overline{x(s)} \mathrm{d}s - b\right) \mathrm{d}a\mathrm{d}b = f(x)(U, t)$$

We note that a more memory efficient representation for $L^2(\mathbb{R}^n)$ and/or a more general representation such as $L^2(\mathrm{SO}(2))$ and $L^2(\mathrm{E}(3))$, have been developed in the context of *steerable* CNNs [33, 64].

## 6  Discussion

### 6.1  Related Works on (G)CNN Universality

**Non-Group CNN.**  Zhou [9, 10] is the earliest to show the *cc*-universality of deep ReLU (non-group) CNNs. In [10], he presented (Theorem 1) the *cc*-universality in $C(\mathbb{R}^d; \mathbb{R})$ in the limit of depth $J \to \infty$, and (Theorem 2) an approximation error rate with respect to $J$. The CNN is carefully designed so that increasing depth also increases width, which is not covered in our GCNN.

**Finite Group CNN.**  Maron et al. [12], Sannai et al. [65], Keriven and Peyré [66], Ravanbakhsh [67] and Petersen and Voigtlaender [13] presented the *cc*-(or $L^p$-)universality results of *finite*-group CNNs. Maron et al. [12] is often cited as one of the earliest publications, where the input space is $\mathcal{X} = \mathbb{R}^{n^k \times a}$ ($a$-channel $k$-th order $n$-dimensional tensors), the output space is $\mathcal{X}' = \mathbb{R}^{n^l \times b}$ ($b$-channel $l$-th order $n$-dimensional tensors), the group $G$ is a subgroup of a symmetric group $\mathrm{S}_n$, and the group action (or representation) $T$ is the left-translation (or left-regular representation). In this setup, they presented the *cc*-universality of deep-ReLU-GCNNs in the space of continuous $G$-equivariant functions $C_{equi}(\mathcal{X}; \mathcal{X}')$. The proofs are indirect because they are based on *invariant polynomials* or *MLPs*. The finite group cases are essentially covered as Example 6 (Deep Sets).

**Lie Group CNN.**  Yarotsky [11] carefully designed deep GCNNs with Lie groups acting on *infinite-dimensional* input/output spaces, and show a version of universality in the space of continuous $G$-equivariant functions $C_{equi}(L^2(G; \mathbb{R}^d); L^2(G; \mathbb{R}^{d'}))$. To be precise, $G$ is either a compact group, translation group $\mathbb{R}^d$, or 2-dimensional roto-translation group $\mathrm{SE}(2)$, and the input/output spaces $\mathcal{X}$ and $\mathcal{X}'$ are square-integrable functions on $G$. The proposed networks are not covered in our GCNNs, but several infinite group cases are covered in Examples 7 and 9.

Remarkably, Kumagai and Sannai [14], Kumagai et al. [15] introduced an integral representation that covers LCH groups, and showed the universality. The proposed integral representation is based on the Haar measure, thus slightly different from ours. The proofs are indirect because the network is converted to an MLP.

### 6.2  Review of Assumptions

**Group $G$.**  We only assume $G$ to be a topological group, to deal with continuous functions on $G$. Thus, a quite large class of groups are covered, for example, all the finite groups such as $\mathbb{Z}_n$ and $\mathrm{S}_n$, compact groups such as $\mathrm{SO}(n)$ and $\mathrm{U}(n)$, and non-compact groups such as $\mathbb{R}^n$ and $\mathrm{E}(n)$ as well.

**Representation Space $\mathcal{X}$.**  Unlike previous studies, it does *not* need to be a function space such as $C(G)$ and $L^2(G/H)$, but it only needs to be an abstract Hilbert space, which is one of the major advantages for geometric understanding of GCNNs. On the other hand, we introduce an auxiliary finite-dimensional subspace $\mathcal{X}_m$ (and projection $P$), to use the Fourier inversion formula on the finite-dimensional Euclidean space $\mathbb{R}^m$ in the proof. We conjecture that the extension to an infinite-dimensional setting would be a routine for some specialists in functional analysis.

**Group Representation $T$.**  It does *not* need to be unitary, irreducible, nor square-integrable, since the proof is based only on a few basic properties of the linear group representation.

**Network Architecture.**  The ridgelet theory supports a wide class of *activation functions*, namely, the tempered distributions ($\mathcal{S}'$). The extension to *deep GCNNs* remains an important open question.

## Acknowledgments and Disclosure of Funding

The authors are grateful to anonymous reviewers for their valuable comments. This work was supported by JST CREST JPMJCR2015 and JPMJCR1913, JST PRESTO JPMJPR2125, and JST ACT-X JPMJAX2004.

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
