# A Proofs

**Additional Notation**   In the proofs, we use two symbols $\widehat{\cdot}$ and $\cdot^\sharp$ for the Fourier transforms in $x \in \mathcal{X}$ and $b \in \mathbb{K}$, respectively. For example,

$$\widehat{f}(\xi) := \int_{\mathcal{X}} f(x) e^{-i\langle \xi, x \rangle_{\mathcal{X}}} \mathrm{d}x, \quad \xi \in \mathcal{X}$$

$$\rho^\sharp(\omega) := \int_{\mathbb{K}} \rho(b) e^{-i\omega b} \mathrm{d}b, \quad \omega \in \mathbb{K}$$

$$\gamma^\sharp(a, \omega) = \int_{\mathbb{K}} \gamma(a, b) e^{-i\omega b} \mathrm{d}b, \quad (a, \omega) \in \mathcal{X} \times \mathbb{K}.$$

With a slight abuse of notation, when $\sigma$ is a tempered distribution (i.e., $\sigma \in \mathcal{S}'(\mathbb{K})$), then $\sigma^\sharp$ is understood as the Fourier transform of distributions. Namely, $\sigma^\sharp$ is another tempered distribution satisfying $\int_{\mathbb{K}} \sigma^\sharp(\omega)\phi(\omega)\mathrm{d}\omega = \int_{\mathbb{K}} \sigma(\omega)\phi^\sharp(\omega)\mathrm{d}\omega$ for any test function $\phi \in \mathcal{S}(\mathbb{K})$.

For any integer $d > 0$ and vector $\boldsymbol{v} \in \mathbb{R}^d$, $|\boldsymbol{v}|$ denotes the Euclidean norm, and $\langle \boldsymbol{v} \rangle := \sqrt{1 + |\boldsymbol{v}|^2}$. For any positive number $t > 0$, $\triangle^{t/2}$ and $\langle \triangle \rangle^t$ denote fractional differential operators defined as Fourier multipliers: for any $\phi \in \mathcal{S}'(\mathbb{R}^d)$,

$$\triangle^{t/2}[\phi](\boldsymbol{v}) := \frac{1}{(2\pi)^d} \int_{\mathbb{R}^d} |\boldsymbol{u}|^t \widehat{\phi}(\boldsymbol{u}) e^{i\boldsymbol{u} \cdot \boldsymbol{v}} \mathrm{d}\boldsymbol{u}, \tag{32}$$

$$\langle \triangle \rangle^{t/2}[\phi](\boldsymbol{v}) := \frac{1}{(2\pi)^d} \int_{\mathbb{R}^d} (1 + |\boldsymbol{u}|^2)^{t/2} \widehat{\phi}(\boldsymbol{u}) e^{i\boldsymbol{u} \cdot \boldsymbol{v}} \mathrm{d}\boldsymbol{u}. \tag{33}$$

In particular when $t = 2$, $\triangle^{t/2}$ coincides with the ordinary Laplacian on $\mathbb{R}^d$.

## A.1   Theorem 1

*Proof.* In the following, we fix a representation $T : G \to GL(\mathcal{X})$ of a group $G$ on a (potentially infinite-dimensional) Hilbert space $\mathcal{X}$ over a field $\mathbb{K}$ equipped with inner product $\langle \cdot, \cdot \rangle_{\mathcal{X}}$, which is a $G$-invariant vector space: $T_G[\mathcal{X}] = \mathcal{X}$, and a finite-dimensional closed subspace $\mathcal{X}_m \subset \mathcal{X}$ equipped with the Lebesgue measure $\lambda$. Let $\mathcal{X}_m^\perp$ be the orthogonal complement of $\mathcal{X}_m$ in $\mathcal{X}$, i.e., $x \oplus y \in \mathcal{X}_m \oplus \mathcal{X}_m^\perp = \mathcal{X}$, and let $P : \mathcal{X} \to \mathcal{X}_m$ denote the orthogonal projection onto $\mathcal{X}_m$. Let $m := \dim_{\mathbb{R}} \mathcal{X}_m$ denotes the real dimension of $\mathcal{X}_m$, and let $k := \dim_{\mathbb{R}} \mathbb{K}$ denotes the real dimension of $\mathbb{K}$, which is either 1 or 2.

Without loss of generality, we can assume (A') that $\gamma(a, \bullet) * \sigma \in \mathcal{S}(\mathbb{K})$ for a.e. $a \in \mathcal{X}_m$, and (A'') that $\gamma^\sharp \sigma^\sharp \in L^1(\mathcal{X}_m \times \mathbb{K})$, which will be eventually justified because later in (38), we set $\gamma^\sharp(a, \omega) = \widehat{f}(\omega a)\overline{\rho^\sharp(\omega)}$.

**Step 1 (Fourier expression).**   Using an identity: For any function $\phi \in \mathcal{S}(\mathbb{K})$ and $b \in \mathbb{K}$, $\phi(b) = \frac{1}{(2\pi)^k} \int_{\mathbb{K}} \phi^\sharp(\omega) e^{ib\omega} \mathrm{d}\omega$, namely the Fourier inversion formula, we can turn $S$ into a *Fourier expression*:

$$S[\gamma](x)(g) = \frac{1}{(2\pi)^k} \int_{\mathcal{X}_m} \int_{\mathbb{K}} \gamma(a, b) \sigma(\langle T_{g^{-1}}[x], a \rangle_{\mathcal{X}} - b) \mathrm{d}b \mathrm{d}a \tag{34}$$

$$= \frac{1}{(2\pi)^k} \int_{\mathcal{X}_m} \int_{\mathbb{K}} \gamma^\sharp(a, \omega) \sigma^\sharp(\omega) \exp\left(i\omega \langle T_{g^{-1}}[x], a \rangle_{\mathcal{X}}\right) \mathrm{d}\omega \mathrm{d}a. \tag{35}$$

By the assumption (A'), the first equation holds at every point $b = (a *_T x)(g)$, and by the assumption (A''), the Fourier expression is *uniformly* absolutely convergent:

$$\int_{\mathcal{X}_m \times \mathbb{K}} |\gamma^\sharp(a, \omega) \sigma^\sharp(\omega) \exp(i\omega(a * x)(g))| \mathrm{d}a \mathrm{d}\omega = \|\gamma^\sharp \sigma^\sharp\|_{L^1(\mathcal{X}_m \times \mathbb{K})} < \infty, \tag{36}$$

for *all* $(x, g) \in \mathcal{X} \times G$. Hence, we can change the order of integration freely.

**Step 2 (Reconstruction).** By changing the variables as $(a, \omega) = (\xi/\omega, \omega)$ with $\mathrm{d}a\mathrm{d}\omega = |\omega|^{-m}\mathrm{d}\xi\mathrm{d}\omega$, we have

$$S[\gamma](x)(g) = \frac{1}{(2\pi)^k} \int_{\mathcal{X}_m \times \mathbb{K}} \gamma^\sharp(\xi/\omega, \omega)\sigma^\sharp(\omega) \exp\left(i\langle T_{g^{-1}}[x], \xi\rangle_{\mathcal{X}}\right) |\omega|^{-m}\mathrm{d}\omega\mathrm{d}\xi. \tag{37}$$

Hence, using a given $f$ satisfying the assumptions (A1) and (A2), and some function $\rho \in \mathcal{S}(\mathbb{K})$, suppose that $\gamma_{f,\rho}$ satisfies the following *separation-of-variables* form:

$$\gamma_{f,\rho}^\sharp(\xi/\omega, \omega) = \widehat{f}(\xi)(e)\overline{\rho^\sharp(\omega)}. \tag{38}$$

Then,

$$S[\gamma_{f,\rho}](x)(g) = \frac{1}{(2\pi)^k} \int_{\mathcal{X}_m \times \mathbb{K}} \widehat{f}(\xi)(e)\overline{\rho^\sharp(\omega)}\sigma^\sharp(\omega) \exp\left(i\langle T_{g^{-1}}[x], \xi\rangle_{\mathcal{X}}\right) |\omega|^{-m}\mathrm{d}\omega\mathrm{d}\xi \tag{39}$$

$$= \left((2\pi)^{m-k} \int_{\mathbb{K}} \sigma^\sharp(\omega)\overline{\rho^\sharp(\omega)}|\omega|^{-m}\mathrm{d}\omega\right)$$

$$\times \left(\frac{1}{(2\pi)^m} \int_{\mathcal{X}_m} \widehat{f}(\xi)(e) \exp\left(i\langle T_{g^{-1}}[x], \xi\rangle_{\mathcal{X}}\right) \mathrm{d}\xi\right) \tag{40}$$

$$\overset{\star}{=} ((\sigma, \rho)) f(PT_{g^{-1}}[x])(e) \tag{41}$$

$$= ((\sigma, \rho)) f(x)(g). \tag{42}$$

where we put

$$((\sigma, \rho)) := (2\pi)^{m-k} \int_{\mathbb{K}} \sigma^\sharp(\omega)\overline{\rho^\sharp(\omega)}|\omega|^{-m}\mathrm{d}\omega. \tag{43}$$

Here, the equality $\overset{\star}{=}$ holds at every continuous point $x_c$ of $f$ for (A2a), in $L^2$ for (A2b), and in $\mathcal{S}'$ for (A2c), respectively.

**Step 3 (Ridgelet transform).** Since we put

$$\gamma_{f,\rho}^\sharp(a, \omega) = \widehat{f}(\omega a)(e)\overline{\rho^\sharp(\omega)}, \tag{44}$$

it is calculated as

$$\gamma_{f,\rho}(a, b) = \frac{1}{(2\pi)^k} \int_{\mathbb{K}} \widehat{f}(\omega a)(e)\overline{\rho^\sharp(\omega)}e^{i\omega b}\mathrm{d}\omega \tag{45}$$

$$= \frac{1}{(2\pi)^k} \int_{\mathbb{K} \times \mathcal{X}_m} f(x)(e)\overline{\rho^\sharp(\omega)}e^{i\omega(b - \langle a, x\rangle_{\mathcal{X}})}\mathrm{d}\omega\mathrm{d}x \tag{46}$$

$$= \int_{\mathcal{X}_m} f(x)(e)\overline{\rho(\langle a, x\rangle_{\mathcal{X}} - b)}\mathrm{d}x, \tag{47}$$

which is the definition of the ridgelet transform for GCNN. $\qquad\square$

### A.2 Theorem 2

*Proof.* Fix arbitrary compact sets $K \subset \mathcal{X}_m$ and $L \subset G$, positive number $\varepsilon > 0$, and function $f \in C_{equi}(K; C(G))$. An $n$-term finite $(G, T)$-CNN is given by

$$f_n(x)(g) := \sum_{i=1}^n c_i\sigma\left((a_i *_T x)(g) - b_i\right), \quad x \in \mathcal{X}_m, \; g \in G \tag{48}$$

with parameters $(a_i, b_i, c_i) \in \mathcal{X}_m \times \mathbb{K} \times \mathbb{C}$. Observe that any finite $(G, T)$-CNN is $(G, T)$-equivariant, that is,

$$f_n(T_g[x])(h) = \sum_{i=1}^n c_i\sigma\left(\langle T_{(g^{-1}h)^{-1}}x, a_i\rangle_{\mathcal{X}} - b_i\right) = f_n(x)(g^{-1}h). \tag{49}$$

Put $\overline{K} := \{T_{g^{-1}}[x] \mid x \in K, g \in L\}$, which is compact because $T$ is continuous, and put $f_e(x) := f(x)(e)$, which is compactly supported, i.e., $f_e \in C(K) \subset C(\overline{K})$. By Theorem 3, there exist a finite number $N \in \mathbb{N}$ and an $N$-term $\mathbb{C}$-valued fully-connected network $F_N(x) = \sum_{i=1}^{N} c_i \sigma(\langle a_i, x\rangle_{\mathcal{X}} - b_i)$ satisfying $\|F_N - f_e\|_{C(\overline{K})} < \varepsilon$. Put $f_N(x)(g) := F_N(T_{g^{-1}}[x])$. Then, it is a $(G, T)$-CNN because

$$f_N(x)(g) = \sum_{i=1}^{N} c_i \sigma\left(\langle T_{g^{-1}}, a_i\rangle_{\mathcal{X}} - b_i\right) = \sum_{i=1}^{N} c_i \sigma\left((a_i *_T x)(g) - b_i\right), \tag{50}$$

and it is an $\varepsilon$-neighbour of $f$ because

$$\|f_N - f\|_{C(K;C(L))} = \sup_{x \in K} \sup_{g \in L} |f_N(x)(g) - f(x)(g)| \tag{51}$$

$$= \sup_{x \in K} \sup_{g \in L} |F_N(T_{g^{-1}}[x]) - f_e(T_{g^{-1}}[x])| \tag{52}$$

$$= \sup_{g \in L} \sup_{x' \in \overline{K}} |F_N(x') - f_e(x')|, \quad x' = T_{g^{-1}}[x] \tag{53}$$

$$< \varepsilon, \tag{54}$$

which concludes the assertion. $\qquad\square$

**Theorem 3** (*cc*-universality of scalar-valued finite fully-connected NNs on $\mathbb{R}^m$). *Suppose that*

1. $\mathcal{X} = \mathcal{X}_m = \mathbb{R}^m$,

2. $f \in C(\mathcal{X}; \mathbb{C})$ *(not vector-valued* $C(\mathcal{X}; \mathbb{C}^G)$ *but scalar-valued), and*

3. *there exists* $k \geqslant 0$ *and* $\theta > 0$ *such that* $\Delta_\theta^k[\sigma] \in L^\infty(\mathbb{R})$ *and Lipschitz continuous.*

*Then, the finite neural networks of the form* $f_n(\boldsymbol{x}) = \sum_{i=1}^{n} c_i \sigma(\boldsymbol{a}_i \cdot \boldsymbol{x} - b_i)$ *are cc-universal, that is, for any compact set* $K \subset \mathbb{R}^m$, *positive number* $\varepsilon > 0$, *and continuous function* $f \in C(K)$, *there exists a finite network* $f_n$ *such that* $\|f - f_n\|_{C(K)} < \varepsilon$.

*Proof.* Since $\sum_{i=1}^{n} c_i \Delta_\theta^k[\sigma](\boldsymbol{a}_i \cdot \boldsymbol{x} - b_i)$ is rewritten as another finite model $\sum_{i=1}^{n'} c_i' \sigma(\boldsymbol{a}_i' \cdot \boldsymbol{x} - b_i')$, it suffice to consider the case $k = 0$. In the following, we assume that $\sigma(= \Delta_\theta^0[\sigma])$ is bounded and Lipschitz continuous.

**Step 1** ($f \sim f_c$). By the density of $C_c^\infty(\mathbb{R}^m)$ in $C(K)$ with respect to the uniform norm, we can take a compactly-supported smooth function $f_c \in C_c^\infty(\mathbb{R}^m)$ satisfying $\|f - f_c\|_{C(K)} < \varepsilon/3$. Since $f_c$ is sufficiently smooth and integrable, there exists a compactly-supported smooth function $\rho \in C_c^\infty(\mathbb{R})$ such that

$$S[R[f_c; \rho]](\boldsymbol{x}) = f_c(\boldsymbol{x}) \text{ at every point } \boldsymbol{x} \in \mathbb{R}^m. \tag{55}$$

For example, take a compactly-supported smooth function $\rho_0 \in C_c^\infty(\mathbb{K})$, write $k = \dim_{\mathbb{R}} \mathbb{K}(= 1 \text{ or } 2)$, and put $\rho(b) := \triangle_b^{m/2}[\rho_0](b) = (2\pi)^{-k} \int_{\mathbb{K}} |\omega|^m \rho_0^\sharp(\omega) e^{ib \cdot \omega} d\omega$. Then, $((\sigma, \rho)) = (2\pi)^{m-k} \int_{\mathbb{K}} \sigma^\sharp(\omega) \overline{\rho^\sharp(\omega)} |\omega|^{-m} d\omega = (2\pi)^{m-k} \int_{\mathbb{K}} \sigma^\sharp(\omega) \overline{\rho_0^\sharp(\omega)} d\omega = (2\pi)^m \int_{\mathbb{K}} \sigma(b) \overline{\rho_0(b)} db = \langle \sigma, \rho_0 \rangle_{L^2(\mathbb{K})}$, which is an ordinary functional inner product, and it is easy to find a $\rho_0$ satisfying $\langle \sigma, \rho_0 \rangle_{L^2(\mathbb{K})} \neq 0$. By normalizing $\rho' := \rho/((\sigma, \rho))$, we can find the $\rho'$. We refer to Sonoda and Murata [54] and Sonoda et al. [45] for more details on the scalar product $((\sigma, \rho))$.

**Step 2** ($R[f_c; \rho]$). To show a discretization $f_n$ of the reconstruction formula converges to $f_c$ in $C(K)$, it is convenient to regard the integration $\int_{\mathbb{R}^m \times \mathbb{R}} [\cdots] d\boldsymbol{a} db$ in $S$ as the Bochner integral, and the integrand $\gamma(\boldsymbol{a}, b) \sigma(\boldsymbol{a} \cdot \boldsymbol{x} - b)$ as a vector-valued function from $\mathbb{R}^m \times \mathbb{R}$ to $C(K)$.

Since $f_c$ is $C^\infty$-smooth, $R[f_c; \rho](\boldsymbol{a}, b)$ is bounded and decays rapidly in $\boldsymbol{a}$, and thus $R[f_c; \rho] \sigma(\boldsymbol{a} \cdot \boldsymbol{x} - b)$ is Bochner integrable, that is,

$$\int_{\mathbb{R}^m \times \mathbb{R}} \sup_{\boldsymbol{x} \in K} |R[f_c; \rho](\boldsymbol{a}, b) \sigma(\boldsymbol{a} \cdot \boldsymbol{x} - b)| d\boldsymbol{a} db < \infty. \tag{56}$$

To see this, the decay property is estimated as follows. For any positive numbers $s, t > 1$,

$$
|R[f_c; \rho](\boldsymbol{a}, b)| = \frac{1}{2\pi} \left| \int_{\mathbb{R}} \widehat{f}_c(\omega \boldsymbol{a}) \overline{\rho^\sharp(\omega)} e^{i\omega b} \mathrm{d}\omega \right|
$$

$$
= \frac{1}{2\pi} \left| \int_{\mathbb{R}} \langle \omega \boldsymbol{a} \rangle^s \langle \omega \boldsymbol{a} \rangle^{-s} \langle b \rangle^t \langle b \rangle^{-t} \widehat{f}_c(\omega \boldsymbol{a}) \overline{\rho^\sharp(\omega)} e^{i\omega b} \mathrm{d}\omega \right|
$$

$$
\leqslant \frac{1}{2\pi} \left| \int_{\mathbb{R}} \langle \omega \boldsymbol{a} \rangle^s \widehat{f}_c(\omega \boldsymbol{a}) \langle \omega \rangle^{-s} \overline{\rho^\sharp(\omega)} \langle \triangle_\omega \rangle^t e^{i\omega b} \mathrm{d}\omega \right| \langle \boldsymbol{a} \rangle^{-s} \langle b \rangle^{-t}, \qquad (57)
$$

which asserts the integrability as below

$$
\int_{\mathbb{R}^m \times \mathbb{R}} \sup_{\boldsymbol{x} \in K} |R[f_c; \rho](\boldsymbol{a}, b) \sigma(\boldsymbol{a} \cdot \boldsymbol{x} - b)| \mathrm{d}\boldsymbol{a} \mathrm{d}b \lesssim \int_{\mathbb{R}^m \times \mathbb{R}} \langle \boldsymbol{a} \rangle^{-s} \langle b \rangle^{-t} \mathrm{d}\boldsymbol{a} \mathrm{d}b < \infty. \qquad (58)
$$

**Step 3 ($f_c \sim f_Q \sim f_n$).** Next, take a compact domain ($m + 1$-dimensional hypercube) $Q := \{(\boldsymbol{a}, b) \in \mathbb{R}^m \times \mathbb{R} \mid |a_i| \leqslant \delta/2, |b| \leqslant \delta/2\}$, and put a band-limited function

$$
f_Q(\boldsymbol{x}) := \int_Q R[f_c; \rho](\boldsymbol{a}, b) \sigma(\boldsymbol{a} \cdot \boldsymbol{x} - b) \mathrm{d}\boldsymbol{a} \mathrm{d}b, \qquad (59)
$$

so that $\|f_c - f_Q\|_{C(K)} < \varepsilon/3$ (by letting $\delta$ sufficiently large). Then, let $Q = \bigsqcup_{i \in I_n} Q_{ni}$ be a decomposition of the domain $Q$ into the union of disjoint family of $|I_n| = n^{m+1}$ cubes with volume $\mathrm{vol}(Q_n) = (\delta/n)^{m+1}$ and the longest diagonal $d_n = \sqrt{m+1}\delta/n$. From each cube, take a point $(\boldsymbol{a}_{ni}, b_{ni}) \in Q_{ni}$ as a center of gravity, that is, so that $c_{ni} = \int_{Q_{ni}} R[f_c; \rho](\boldsymbol{a}, b) \mathrm{d}\boldsymbol{a} \mathrm{d}b = R[f_c; \rho](\boldsymbol{a}_{ni}, b_{ni}) \mathrm{vol}(Q_n)$, and put $w_{ni} := R[f_c; \rho](\boldsymbol{a}_{ni}, b_{ni})$, then put a finite network as

$$
f_n(\boldsymbol{x}) := \sum_{i \in I_n} c_{ni} \sigma(\boldsymbol{a}_{ni} \cdot \boldsymbol{x} - b_{ni}). \qquad (60)
$$

**Step 4 ($f_Q \sim f_n$).** We show $f_n \to f_Q$ in $C(K)$. First, the integrands converge to the limit at almost every $(\boldsymbol{a}, b) \in Q_{ni}$ as

$$
\sup_{\boldsymbol{x} \in K} \left| R[f_c; \rho](\boldsymbol{a}, b) \sigma(\boldsymbol{a} \cdot \boldsymbol{x} - b) - w_{ni} \sigma(\boldsymbol{a}_{ni} \cdot \boldsymbol{x} - b_{ni}) \right| \qquad (61)
$$

$$
\leqslant \sup_{\boldsymbol{x} \in K} \left| R[f_c; \rho](\boldsymbol{a}, b) \right| \left| \sigma(\boldsymbol{a} \cdot \boldsymbol{x} - b) - \sigma(\boldsymbol{a}_{ni} \cdot \boldsymbol{x} - b_{ni}) \right|
$$

$$
+ \left| R[f_c; \rho](\boldsymbol{a}_{ni}, b_{ni}) - R[f_c; \rho](\boldsymbol{a}, b) \right| \left| \sigma(\boldsymbol{a}_{ni} \cdot \boldsymbol{x} - b_{ni}) \right| \qquad (62)
$$

$$
\leqslant \|R[f_c; \rho]\|_\infty \mathrm{Lip}(\sigma) \sup_{\boldsymbol{x} \in K} \left| (\boldsymbol{a} - \boldsymbol{a}_{ni}) \cdot \boldsymbol{x} - (b - b_{ni}) \right|
$$

$$
+ \mathrm{Lip}(R[f_c; \rho]) \left| (\boldsymbol{a} - \boldsymbol{a}_{ni}, b - b_{ni}) \right| \|\sigma\|_{L^\infty(\mathbb{R})} = O(\delta/n) \to 0 \quad n \to \infty. \qquad (63)
$$

Besides, the integrands are uniformly bounded as

$$
\sup_{\boldsymbol{x} \in K} \left| w_{ni} \sigma(\boldsymbol{a}_{ni} \cdot \boldsymbol{x} - b_{ni}) \right| \leqslant \left| R[f_c; \rho](\boldsymbol{a}, b) \right| \|\sigma\|_{L^\infty(\mathbb{R})}, \text{ for a.e. } (\boldsymbol{a}, b) \in Q_{ni}. \qquad (64)
$$

Therefore, by the dominated convergence theorem for the Bochner integral, we have

$$
\|f_Q - f_n\|_{C(K)} = \sup_{\boldsymbol{x} \in K} \left| \sum_{i \in I_n} \int_{Q_{ni}} R[f_c; \rho](\boldsymbol{a}, b) \sigma(\boldsymbol{a} \cdot \boldsymbol{x} - b) \mathrm{d}\boldsymbol{a} \mathrm{d}b - \sum_{i \in I_n} c_{ni} \sigma(\boldsymbol{a}_{ni} \cdot \boldsymbol{x} - b_{ni}) \right|
$$

$$
(65)
$$

$$
\leqslant \sum_{i \in I_n} \int_{Q_{ni}} \sup_{\boldsymbol{x} \in K} \left| R[f_c; \rho](\boldsymbol{a}, b) \sigma(\boldsymbol{a} \cdot \boldsymbol{x} - b) - w_{ni} \sigma(\boldsymbol{a}_{ni} \cdot \boldsymbol{x} - b_{ni}) \right| \mathrm{d}\boldsymbol{a} \mathrm{d}b \qquad (66)
$$

$$
\to 0, \quad n \to \infty. \qquad (67)
$$

Hence by letting $n$ sufficiently large, we have $\|f_n - f_Q\|_{C(K)} < \varepsilon/3$.

To sum up, we have shonw the *cc*-universality:

$$
\|f - f_n\|_{C(K)} \leqslant \|f - f_c\|_{C(K)} + \|f_c - f_Q\|_{C(K)} + \|f_Q - f_n\|_{C(K)} < \varepsilon. \qquad (68)
$$

$\square$

**Notes.** In the proof, we employed a naive discretization based on the regular grids in $Q$. However, since we know the closed-form expression of the ridgelet transform, we can discretize it more effectively. For example, a better discretization scheme is investigated in the so-called *Maurey-Jones-Barron (MJB) theory* and the dimension independent *Barron's bound* ([see, e.g., 68]).