# OpenReview forum: "Universality of Group Convolutional Neural Networks Based on Ridgelet Analysis on Groups"
_NeurIPS.cc/2022/Conference — NeurIPS 2022 Accept_

### Official Review · Reviewer_8Jwn · 2022-07-10

**Rating:** 5
**Confidence:** 2
**Soundness:** 4 excellent
**Presentation:** 4 excellent
**Contribution:** 2 fair

**Summary:**

The authors prove the (cc-) universality property of group convolutional neural networks (GCNNs) with one hidden layer using the tools from ridglet theory.
To do so, the authors derive the first closed form solution for the ridglet transform of GCNNs, covering a wide spectrum of existing equivariant model designs.


**Questions:**

The authors claim that existing universality proofs for GCNNs are indirect and rely on the universality of either fully-connected networks or invariant polynomials, while the proposed method is direct and constructive.
Unfortunately, I am not familiar with the literature on the universality property; could the authors elaborate further about the limitations of these existing methods and the benefit of the proposed one?

Is this proof considered constructive because of the existence of the reconstruction formula for the ridglet transform?

What kind of conclusions can be drawn from this theoretical work?
Does this proof provide any new intuition about GCNNs with respect to previous methods?


Line 189: why is this an advantage for geometric understanding of CNNs? Could you elaborate further on this?


**Ethics Review Area:**

["I don’t know"]

**Limitations:**

While the work seems rigorous, I think it misses some motivation and an analysis of the theoretical results.
However, I am not familiar with the related literature on universality properties of neural networks.


**Strengths And Weaknesses:**

The manuscript is well written and the presentation is clear.
The contributions are clear and novel and the theoretical analysis seem rigorous (although I am not an expert on ridglet analysis and universality proofs).

While the universality of GCNNs is a well known fact, existing proofs were only indirect or existential, while the proposed proof is constructive.
However, I feel like the manuscript does not motivate sufficiently well why this difference is important; see "Questions".

Finally, while the work seems quite rigorous, the authors do not comment much on their final result.
Indeed, the paper is missing a proper "conclusion" section, stating what the main take-away points should be.

---

> ### Author Response · Authors · 2022-08-02
> **Reply to Reviewer 8Jwn**
>
> Q. The authors claim that existing universality proofs for GCNNs are indirect and rely on the universality of either fully-connected networks or invariant polynomials, while the proposed method is direct and constructive. Unfortunately, I am not familiar with the literature on the universality property; Could the authors elaborate further about the limitations of these existing methods and the benefit of the proposed one?
>
> Yes. Please refer to the third response to Reviewer onTd.
>
> Q. Is this proof considered constructive because of the existence of the reconstruction formula for the ridgelet transform?
>
> Yes. As long as H is finite-dimensional and all the information of f is available, then we can compute both R[f] and S[R[f]] by using any numerical integration algorithm. In the course of computation, we obtain a series of finite networks f_n, where n corresponds to the iteration number in the numerical integration algorithm (see e.g. Step 3 in the proof of Theorem 2). As a consequence of the numerical integration, f_n converges to S[R[f]] as n \to \infty, and the construction of f_n is indicated by the numerical integration algorithm, thus the proof of universality by discretizing-the-reconstruction-formula becomes automatically constructive.
>
> As detailed in the third response to Reviewer onTd, conventional proofs are often existential because, for example, they rely on the existential theorems such as the Stone-Weierstrass theorem, which only states the existence of polynomials without the construction. (Recall that the Taylor/Fourier series expansion is not the one given in the Stone-Weierstrass theorem.)
>
> Q. What kind of conclusions can be drawn from this theoretical work? Does this proof provide any new intuition about GCNNs with respect to previous methods?
>
> Some previous studies have also presented a construction of parameters, (which is often a case-by-case construction,) but these are simply “sufficient conditions” for a network to represent a given function f. For FNNs, the ridgelet transform has been shown to be the minimizer of the regularized least-squares problem (see [41]), and thus the ridgelet transform is key to analyze solutions of deep learning problems. For CNNs, the parallel claim is not yet shown, but we expect that the ridgelet transform will be key as well.
>
> Q. Line 189: why is this an advantage for geometric understanding of CNNs? Could you elaborate further on this?
>
> There are several formalisms of “geometric understanding” (just like there are several branches in geometric deep learning and information geometry). Here, we use “geometric” in the sense that the GCNN S and the ridgelet transform R are defined in a coordinate-free manner. As presented in the Examples section, instantiations of S and R (or H,G, and T) vary a great deal. Nevertheless, the reconstruction formula S[R[f]]=f and the universality hold for each individual instantiation. In fact, these are major benefits of the “geometric” formulation, and we can “geometrically understand” why and how Theorem 1 (S[R[f]]=f) holds in the proof: As long as Steps 1 and 2 (the Fourier expression) pass, then we can systematically find the R as a separation-of-variables form. And in order for Steps 1 and 2 to pass, we can see that “geometric” assumptions such as “H is a Hilbert space” and “T is a group homomorphism” are essential.
>
> On the contrary, conventional proofs have been shown case-by-case. We consider this as a result of less geometric formulations of CNNs.

---

### Official Review · Reviewer_sVAk · 2022-07-11

**Rating:** 7
**Confidence:** 2
**Soundness:** 4 excellent
**Presentation:** 2 fair
**Contribution:** 4 excellent

**Summary:**

[updated score from 6->7 based on technical soundness and contribution to DL theory]

The paper describes a mathematical analysis of group convolutional neural networks, showing that they are universal approximators. To show this, the authors derived the ridgelet transform (a mathematical analysis tool) for G-CNNs. This is a result in itself, which is further used in this paper to provide a constructive proof of universality.

**Questions:**

Important:
1. Why is it important to derive the ridgelet transform?
2. How should I interpret this transform?
3. Why can we show universality once it is derived?

Less important:

4. Line 87: "As an application, we show ..." Are there any other applications of the ridgelet transform? I find it hard to assess the impact of the result of deriving it.

5. Line 184: Fourier transform on $\mathcal{H}$, this is way more abstract then I have ever seen the FT (most general to me is FT based on group irreps). Is it possible to give interpretation to this? Why do we still call it a Fourier transform?

**Limitations:**

I see no issues here.

**Strengths And Weaknesses:**

**Strengths**
* The paper is technically sound and rigorous
* The proposed proof constructive and does not rely on indirect proofing methods

**Weaknesses**
* The paper is very technical, and could perhaps better fit a mathematical functional analysis venue, rather than an ML conference. The theme (universality of G-CNNs) falls within the conference scope, but the content otherwise is purely mathematical
* Related is my concern about the impact of the paper in the field. What is the relevance of a new universality proof? I see the beauty/elegance of the paper, but that is on math merits, not on ML principles per se.
* The paper could overall benefit from a more intuitive exposition. Why is it important to derive the ridgelet transform? How should I interpret this transform? Why can we show universality once it is derived?

I believe the paper could be a great contribution to NeurIPs if it were to be made more accessible to a broader audience (like me, I like to learn from these papers, but this one was actually far out my comfort zone). E.g. up and until page 3.2 I could clearly follow everything, but what comes after I had to read several times to understand, and still don't fully understand the significance of e.g., 3.3. These are things a mathematician may get straight away though..

**Minor details**

line 54: Could you provide some intuition on the role of $\gamma$ in equation 2. It somehow weights the different parameter sets (kernel a, bias b) and can be used to rewrite into (1), but a bit more intuition may be helpful.

line 64: What does $C(G)$ denote? Only later are such spaces defined, but not yet this early..

Line 69: I like the analysis - synthesis viewpoint of R and S, in relation to the Fourier transform. This was helpful.

Line 87: "As an application, we show ..." Are there any other applications of the ridgelet transform? I find it hard to assess the impact of the result of deriving it.

Line 126: Why do you refer to it as a *generalized* form of the group convolution? I have seen this form several times with $\mathcal{H}=\mathbb{L}_2(G/H)$. Does generalized refer to the fact that $\mathcal{H}$ can be different then the usual function spaces?

Line 151: This sentence misses a part!

Line 158: Why does f now map to $\mathbb{C}^G$ and not $C(G)$?

Line 184: Fourier transform on $\mathcal{H}$, this is way more abstract then I have ever seen the FT (most general to me is FT based on group irreps). Is it possible to give interpretation to this? Why do we still call it a Fourier transform?

Line 221: What does notation $f(\cdot,e)$ mean? Is it the same as $f(\cdot)(e)$?

Line 263: This is another example of where I understand the explicit forms (of the ridgelet trafo and reconstruction formula), however, I do not understand why it is relevant for me to read this. One the one hand it is nice to see explicit examples, on the other hand side I do not understand what to do with it? E.g., what are the implications to the universality proof?

---

> ### Author Response · Authors · 2022-08-02
> **Reply to Reviewer sVAk**
>
> Q. What is the relevance of a new universality proof? I see the beauty/elegance of the paper, but that is on math merits, not on ML principles per se.
> Q. Why is it important to derive the ridgelet transform?
>
> The reparametrization trick of parameters (a,b) as distributions \gamma is a common practice in today's deep learning theory (e.g., NTK, lazy learning, lottery tickets, mean field theory, Langevin dynamics). For example, it has been pointed out both theoretically and empirically that the final distribution of parameters is close to the initial distribution (namely a prior such as the Gaussian distribution). This is considered as an implicit regularization in deep learning. Then, what function is represented by a parameter distribution that is close to the initial distribution? Previous studies on universality have also provided several constructions of parameters, but these are simply “sufficient conditions” for a network to represent a given function f. For FNNs, on the other hand, the ridgelet transform has been shown to be the minimizer of the least-squares problem (see [41]), and thus the ridgelet transform is key to the analysis of the question. For CNNs, the claim is not yet strictly shown, but we can expect that the ridgelet transform is key as well. (We note that other theories such as NTK and Gibbs distribution can also provide the final distribution, but they are more limited or indirect than the ridgelet transform.)
>
> Q. How should I interpret this transform?
>
> The ridgelet transform R[f;\rho] is a bilinear map in f and \rho. As briefly mentioned in the Introduction, R and S play the parallel roles as F and F^{-1}. It would be interesting to check if parallel formulas hold such as for translation, scaling, multiplication, differentiation, convolution, Plancherel, … .
>
> Q. Why can we show universality once it is derived?
>
> Because a finite network is a Riemannian sum (numerical integration scheme) of the integral representation network.
>
> (minor comments)
>
> Q. ... up and until page 3.2 I could clearly follow everything, but what comes after I had to read several times to understand, and still don't fully understand the significance of e.g., 3.3.
>
> In Section 3.3, a function norm and a function class for vector-valued functions are defined. We use them in Theorem 2 to show that GCNNs can approximate any functions in this function space with respect to this function norm.
>
> In Section 3.4, a measure \lambda on an abstract Hilbert space H, and the Fourier transform on H are defined. We use \lambda to define the integral representation in Definition 3, and the Fourier transform in the proof of Theorem 1.
>
> Q. line 54: Could you provide some intuition on the role of \gamma in equation 2. It somehow weights the different parameter sets (kernel a, bias b) and can be used to rewrite into (1), but a bit more intuition may be helpful.
>
> Your understanding is quite correct. If S is not a neural network but a Fourier transform, \gamma is the distribution of frequencies. If the intensity |\gamma(a,b)| is high at some (a0,b0), then the network likely contains the “frequency (a0,b0)” or a “feature \sigma( (a0 * x) - b0 )”.
>
> Q. Line 87: "As an application, we show ..." Are there any other applications of the ridgelet transform? I find it hard to assess the impact of the result of deriving it.
>
> In addition to the applications in conventional deep learning theory, please refer to the second final paragraph in the Introduction (and references such as Savarese et al. [36] and Sonoda et al. [41]).
>
> Q. Line 126: Why do you refer to it as a generalized form of the group convolution? I have seen this form several times with H=L2(G/H) . Does generalized refer to the fact that H can be different then the usual function spaces?
>
> Right. H is an abstract Hilbert space and need not be L^2 function spaces such as L^2(G) and L^2(G/H). Note that L^2(G) (in a standard textbook) can be defined only when G is an LCH group. In Definition 1, we define a generalized “group convolution” as a “matrix element”, which includes the standard group convolution as shown in Eq.11.
>
> Q. Line 158: Why does f now map to C^G and not C(G)?
>
> Because continuity is not necessary to define the equivariance.
>
> Q. Line 184: Fourier transform on H, this is way more abstract then I have ever seen the FT (most general to me is FT based on group irreps). Is it possible to give interpretation to this? Why do we still call it a Fourier transform?
>
> We simply identify a finite-dimensional H as a Euclidean space R^n. Of course, it depends on the choice of orthonormal basis {e_i}. Example 5 (periodic convolution) is simple but non-trivial. Namely, the choice of basis corresponds to the limitation of frequencies.
>
> Q. Line 221: What does notation f(.,e) mean? Is it the same as f(.)(e)?
>
> Yes. It’s a typo used in the old version.

---

### Official Review · Reviewer_onTd · 2022-07-13

**Rating:** 6
**Confidence:** 2
**Soundness:** 3 good
**Presentation:** 2 fair
**Contribution:** 2 fair

**Summary:**

This paper uses the ridgelet theory technique to show the universality of group convolutional neural networks for various groups and feature spaces. Their results are new in the way that they apply ridgelet theory to group CNNs. Prior works have used ridgelet transform to show the universality of fully-connected neural networks, and literature exists on the universality of GCNN for limited groups.

**Questions:**

Answering the following questions is to provide insight into the significance of the method and its novelty. Please explain

1. How the method differs from others?
2. How difficult and unique the analysis are compared to the one using ridgelet for fully-connected networks? Is it a trivial extension?
3. How do their universality results for GCNN differ from already existing literature on GCNN? The authors have briefly touched on this, but it needs more clarification.


**Strengths And Weaknesses:**

The universality of GCNN is of interest to the community. The paper provides a proper amount of details and explanations. However, I found the paper a bit hard to follow. Many terms are defined without additional explanation/elaboration. For example, Section 5 provides several examples without any description. Moreover, I recommend covering related works in more depth.

The originality and significance of the work are unclear: 1) ridgelet transform as a technique is previously used for the universality of fully-connected networks, and 2) prior works have shown the universality of GCNN under certain conditions.

---

> ### Author Response · Authors · 2022-08-02
> **Reply to Reviewer onTd**
>
> 1. How does the method differ from others?
>
> Our proof is more unified and constructive than previous proofs. Some studies have also presented constructive proofs. However, they all construct the parameters/architectures for each individual network design. These are case-by-case, and if the network design changes, the construction method has to be redesigned from scratch. By discretizing the ridgelet transform, we can automatically generate the construction method (at least) as many as the variety of numerical integration methods. Hence, the ridgelet transform induces construction methods. It is an interesting open question to check the inverse “every construction method is induced as a discretization scheme of ridgelet transforms”. For fully-connected NNs, Sonoda et al. [42] have recently given an affirmative answer.
>
> 2. How difficult and unique the analysis is compared to the one using ridgelet for fully-connected networks? Is it a trivial extension?
>
> There is no unique way to formulate the "integral representation of CNNs," and we spent three years developing various prototypes. For example, since the convolution on the “Euclidean space" can be written using Töplitz matrices, one could consider a formulation such as \int \gamma(A,b) \sigma(Ax-b) dA db where the parameter A is a matrix, but this leads to a completely different ridgelet transform (related to the d-plane transform), and can cover less symmetry, since it depends on the Euclidean coordinates.
>
> Note that some studies claim the “equivalence of CNNs and FNNs,” which is simply misleading because the CNNs and FNNs in such studies are very carefully designed. In general, FNNs and CNNs are defined on completely different spaces, namely Euclidean spaces and function spaces, and there is no canonical way to identify them.
>
> 3. How do their universality results for GCNN differ from already existing literature on GCNN? The authors have briefly touched on this, but it needs more clarification.
>
> Zhou (2018,2020) [11,12].
> This seems to be the earliest result. Zhou presented (in Theorem 1) the cc-universality in C(R^d;R) in the limit of infinite depth J, and (in Theorem 2) an approximation error rate wrt J. The CNN is carefully designed so that increasing the depth results in increasing the width. The activation function is limited to ReLU, which is essential for the proof. Since it is different from standard CNNs (as well as group CNNs), the “universality of CNN” in this paper is at a concept level. Specifically, our result does not directly cover their result.
>
> Maron et al. (2019) [14]. (in addition, Sannai-Takai-Matthieu, arXiv:1903.01939; Keriven-Peyré, NeurIPS2019; Ravanbakhsh, NeurIPS2020; Petersen and Voigtlaender (2020) [15]).
> These are cc- (or L^p-) universality results of “finite-group” CNNs. Maron et al. (2019) is often cited as one of the earliest publications, where the input space is H=R^{n^k \times a} (a-channel k-th order n-dim. tensors), the output space is H’=R^{n^l \times b} (b-channel l-th order n-dim. tensors), the group G is a subgroup of symmetric group S_n, and the group action is the left-translation (i.e. the group representation T is the left-regular representation). In this setup, they presented the cc-universality of deep-ReLU-GCNNs in the space of continuous G-equivariant functions C_{equi}(H;H’). The proof is constructive, but still existential, because it is based on the existence of invariant polynomials (a version of Stone-Weierstrass), which does claim the existence but does not explain the construction. In other words, we cannot determine the coefficients and degrees of those polynomials. These finite-group cases are essentially covered as Example 4 (Deep Sets). Again, the proof based on the ridgelet transform can further provide the construction of parameters, and thus be more constructive.
>
> Yarotsky (2021) [13].
> This is a journal paper composed of three major sections. Section 2 discusses the cc-universality of depth-2 (= shallow) "FNNs" in the space of G-equivariant continuous functions C_{equi}(R^m;R^n), when G is compact (note: finite <= compact <= non-compact). Sections 3 and 4 show the cc-universalities of “carefully designed” deep GCNNs in the space of G-equivariant continuous functions C_{equi}( L^2(G;R^d) ; L^2(G;R^d’)) where G is either translation group R^d and 2-dim. roto-translation group SE(2). This study aims to investigate a new convolution-like structure that explicitly encodes the group-equivariance structure as explicit inductive bias, particularly when the input/output feature spaces are infinite-dimensional spaces (i.e. L^2(G:R^d) ) and the group G is an infinite group. (We note that eventually the input/output spaces are finite-dimensional since the spaces are discretized as finite-dimensional Euclidean spaces). Thus, the proposed networks are not covered in our study. However, the cases of infinite-dimensional groups acting on a function space are covered in Examples 5 and 7.

---

> > ### Comment · Reviewer_onTd · 2022-08-04
> > **After Reading Authors' Responses**
> >
> > The authors have addressed most of my comments. Including the authors' comment in the manuscript improves the motivation and related works. I have updated my score accordingly.

---

### Official Review · Reviewer_VpiC · 2022-07-16

**Rating:** 5
**Confidence:** 4
**Soundness:** 3 good
**Presentation:** 3 good
**Contribution:** 3 good

**Summary:**

This paper considered the approximation property of GCNNs on the frame of the ridgelet theory. By representing a typical GCNN into a nonlinear mapping between group representations, corresponding formulations on $(G,T)-convolution$ and $(G,T)-equivariance$ are established for the convenience of further demonstration under ridgelet-frame. The main results showed the cc-universality of GCNNs in a constructive and unified manner.

**Questions:**

Is graph equivariant NNs (such as SE(3)-transformer) an example of ridgelet transform-based CNNs?

**Limitations:**

This is theoretical study. There is no negative societal impact.

**Strengths And Weaknesses:**

*Strengths: The ridgelet transform induced group convolution can cover typical class of neural network convolutions such as deep sets and E(n)-equivariant convolutions. This study derives the ridgelet transform for a general class of GCNNs. The reconstructed formula is given which discretized version is then applied to prove the universality of (G,T)-CNNs. The cc-universality of GCNNs for a general class of group equivalent continuous vector-valued functions is shown as an example. Examples of GCNNs are given.
*Weaknesses: It is best the numerical experiments are shown to test the theory developed.

---

> ### Author Response · Authors · 2022-08-02
> **Reply to Reviewer VpiC.**
>
> Q. Is graph equivariant NNs (such as SE(3)-transformer) an example of ridgelet transform-based CNNs?
>
>
> We have skimmed the SE(3)-transformer paper (NeurIPS 2020 version), and the answer is “no”, but it would be interesting to consider why and how “no”.
>
> Following the arguments of Tensor Field Networks (TFNs), an SE(3)-equivariant graph NN is an SE(3)-equivariant mapping that eats a d-dimensional vector-field on R^3, say z_in=(x,f_in), and spits out an r-dimensional vector-field on R^3, say z_out = (x,f_out). (When z_in and z_out are sums of n point-masses respectively, then they are strictly equivalent to vector-valued features on an n-vertices graph, say \Gamma).
> Let H_in and H_out be any Hilbert spaces of input/output vector-fields z_in and z_out, respectively, and let T be any group representation of SE(3) on H_in. That is, T is a group homomorphism from SE(3) to GL(H_in). In the SE(3)-transformer paper, T is supposed to be the regular representation.
> Our main theorem states that any (SE(3),T)-equivariant continuous map \phi : H_in \to C(SE(3)) can be expressed as a (SE(3),T)-CNN,
>
> C(z_in)(g) = \int_{H_in \times R} \gamma(a,b) \sigma( (a * z_in)(g) - b ) d\lambda(a) db,
>
> (and the ridgelet transform \gamma = R[\phi] provides a closed-form description of a particular \gamma). While in SE(3)-transformer paper, following the argument by Wang et al, an SE(3)-transformer computes an SE(3)-equivariant mapping A : H_in \to H_out with
>
> A(z_in)(x) = \int_{H_in} w( z_in, z ) h(z)(x) d \mu(z),
>
> where w : H_in \times H_in \to R is an attention weight function, h : H_in \to H_out is a value-embedding, and \mu is an appropriate measure on H_in. While both maps are linear transformations of feature maps on H_in, namely \sigma(...z_in…) and h(z_in), a mathematical difference is that while the weight function \gamma in C is a function of parameters (a,b), w in A is a function of inputs z_in.

---

### Author Response · Authors · 2022-08-02
**General Comments**

We would like to thank all the reviewers for their constructive feedback and the time spent reading the paper.

In reference to the reviewers' concerns, we have updated Section 1 (Introduction) and Section 6 (Related Work) in the revised version.

---

### Meta-Review · Area_Chair_SgJw · 2022-09-01

**Recommendation:** Accept
**Confidence:** Less certain

**Metareview:**

The paper studies the approximation properties of group convolutional neural networks. It establishes the “cc-universality” of group CNNs, i.e. that such networks can approximate any continuous function over any compact set, using a new constructive approach which is based on a generalization of the ridgelet transform. The proof is constructive, in the sense that approximating networks are given in closed form by discretizing the transform. This approach may have applications beyond the scope of the paper — most immediately, to identifying classes of functions for which neural network approximations are accurate in a quantitative sense; this can be tied to the decay properties of the ridgelet transform. Reviewers found the paper to be clearly written and of high technical quality, albeit somewhat mathematically dense in its presentation. Universal approximation theorems provide an important piece of theoretical background, as well as a “sanity check” for new network architectures; having a unified, constructive approach to derive them could stimulate further work.

**Award:**

No

---

### Decision · Program_Chairs · 2022-09-14

Accept